# Joint profiling of cell morphology and gene expression during in vitro neurodevelopment

Adithi Sundaresh[1†], Dimitri Meistermann[1†], Riina Lampela[1], Zhiyu Yang[2], Rosa Woldegebriel[1], Andrea Ganna[1,2,3], Pau Puigdevall Costa[1], Helena Kilpinen[1,2,3,4]*

[1]Helsinki Institute of Life Science (HiLIFE), University of Helsinki, Helsinki, Finland; [2]Institute for Molecular Medicine Finland (FIMM), University of Helsinki, Helsinki, Finland; [3]Faculty of Medicine, University of Helsinki, Helsinki, Finland; [4]Faculty of Biological and Environmental Sciences, University of Helsinki, Helsinki, Finland

**\*For correspondence:**
helena.kilpinen@helsinki.fi

[†]These authors contributed equally to this work

**Competing interest:** The authors declare that no competing interests exist.

**Abstract** Differentiation of induced pluripotent stem cells (iPSCs) toward neuronal lineages has enabled diverse cellular models of human neurodevelopment and related disorders. Here, we jointly profiled neuronal morphology and gene expression at single-cell resolution across 60,000 iPSC-derived cortical neurons at three developmental time points with Cell Painting (CP) and single-cell RNA-sequencing (scRNA-seq). By modeling the relationship between morphological features and gene expression within our differentiation system, we annotated image-based features with biological functions and showed that while CP resolves broader neuronal classes than scRNA-seq, it complements transcriptomic data by quantifying the biological processes that drive neuronal differentiation over time, such as mitochondrial function and cell cycle. Further, we found that while over 60% of the cells resembled those seen in the fetal brain, 28% represented metabolically abnormal cell states and broader neuronal classes specific to *in vitro* cells. We show that iPSC-derived cortical neurons are nonetheless a relevant model for a range of brain-related complex traits, including schizophrenia and bipolar disorder, and that disease heritability can also be captured in the morphological feature space. Finally, we applied CP to iPSC-derived neural progenitors from patients with Kabuki syndrome, revealing morphological signatures of precocious differentiation and altered cell cycling. These results highlight the potential of multi-modal single-cell characterization to reveal complementary and disease-relevant cellular and molecular phenotypes.

## Editor's evaluation

This study presents a valuable description of human iPSC-derived neuron differentiation and highlights the utility of this cell model for investigating neurodevelopmental traits and diseases. The use of paired morphological and transcriptomic data is compelling, though the morphological data is limited in resolution and cannot distinguish cell subtypes.

## Introduction

Induced pluripotent stem cells (iPSCs) have enabled cell-level profiling experiments in many previously inaccessible cell types and lineages. For example, human iPSC-derived neurons and glia have revolutionized the study of brain-related disorders, which previously relied heavily on animal models and post-mortem tissue. However, iPSC-based differentiation systems are inherently variable (*Jerber et al., 2021*; *Kilpinen et al., 2017*; *Volpato et al., 2018*) and for any given protocol, the entire

diversity of cell types generated *in vitro* is not fully characterized. For example, *in vitro* conditions can give rise to cell types and states that are not seen *in vivo* (*Bhaduri et al., 2020*). Single-cell RNA-sequencing (scRNA-seq) has transformed the resolution at which iPSC-derived cell types can be characterized. However, gene expression levels alone do not comprehensively reflect changes in cellular function and the biological processes that ultimately drive disease pathophysiology (*El-Brolosy and Stainier, 2017*), highlighting the need to measure additional cellular phenotypes.

In this study, we explored the utility of Cell Painting (CP), a high-content image-based assay, in expanding the spectrum of phenotypic readouts available from single cells. We performed joint profiling of cellular morphology (CP) and gene expression (scRNA-seq) in iPSC-derived cortical neurons. CP uses fluorescent dyes to label different basic organelles of the cell, such as the nucleus, mitochondria, and the endoplasmic reticulum (ER) from which hundreds of image-based features can be derived, representing the morphological profile of each cell (*Gustafsdottir et al., 2013*; *Bray et al., 2016*). Compared with other imaging technologies, CP is more affordable and transferable across cell types. It has been used extensively for large-scale screening experiments, for e.g. in the context of drugs (*Chandrasekaran et al., 2024*; *Haghighi et al., 2022*; *Chandrasekaran et al., 2023*), and therefore is highly suitable for the unbiased characterization of *in vitro* systems where phenotypes are not known *a priori*.

We applied CP to an established cortical neuron differentiation system based on dual-SMAD inhibition, chosen for its reported capacity to recapitulate the progression of neurodevelopment *in vitro* (*Shi et al., 2012a*; *Shi et al., 2012b*). We collected morphological data from 54,415 cells from four healthy donors across three time points corresponding to early progenitors (day 20), intermediate progenitors (IPs, day 40), and maturing cortical neurons (day 70) (*Figure 1a*). Cell type heterogeneity in this system, and in the brain more generally, is high, and our aim was to test to what degree CP is able to disentangle closely related cell types and what phenotypic information is gained beyond transcriptomic data. To establish a cell type reference, we also collected single-cell transcriptomic data from 62,929 cells at the same three developmental time points across the differentiation and compared them to fetal cell types *in vivo* (*Figure 2a*). CP data is typically analyzed at a bulk- (well) level, and systematic interpretation of image-derived features, directly or in the context of other data modalities, is only beginning to be explored (*Caicedo et al., 2022*). Here, we analyzed the CP data at a single-cell resolution and, by leveraging a predictive model (*Haghighi et al., 2022*), provided links between image features and gene expression levels in developing cortical neurons (*Figure 2b*).

Over 60% of the cells in our cortical system reliably matched cell types in the fetal brain, while an additional 28% represented broader neuronal classes observed in stem cell-derived organoids and metabolically abnormal cell states. We found that CP was able to resolve progenitor versus neuronal cell classes, and donor-specific changes were captured by both assays. Image-based features were informative in quantifying the dynamic nature of cellular differentiation, as observed with changes in mitochondrial intensity and cell cycle over time. In addition, the joint analysis of cell morphology features and gene expression allowed us to link these dynamic processes with known cell biology (*Nassiri and McCall, 2018*).

Finally, we evaluated the disease relevance of *in vitro* differentiated cells using stratified linkage disequilibrium (LD) score regression (*Finucane et al., 2018*), in both the transcriptomic and morphological spaces. Overall, iPSC-derived cells captured a significant fraction of heritability of brain-related traits including schizophrenia, ADHD, and bipolar disorder, based on their cell-type-specific marker genes or genes predictive of marker features in CP. Given the ability to capture disease heritability also in the morphological space, we profiled an additional 11,974 in vitro differentiated neural progenitors with CP, including iPSC lines derived from individuals with Kabuki syndrome (KS), a rare neurodevelopmental disorder (NDD). By applying the predictive model trained on healthy donors, we identified cellular phenotypes related to premature differentiation and reduced cell cycling in KS cells. We provide proof of concept that CP can be used to identify disease-relevant phenotypes and present an approach to integrate transcriptomic and image-based phenotypes from single cells for downstream disease modeling studies.

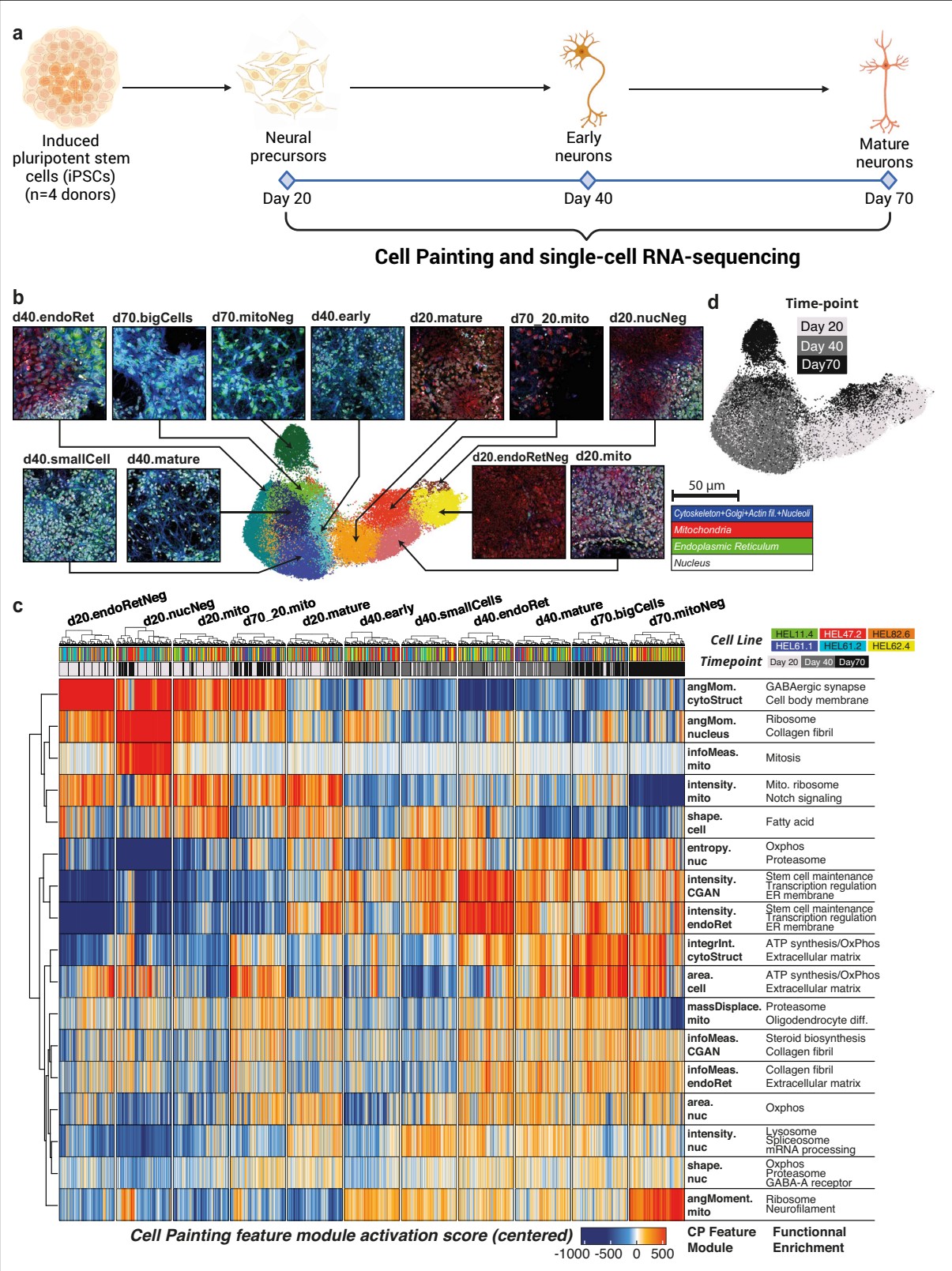

**Figure 1.** Overview of the experimental design and Cell Painting (CP) at the single-cell level. (**a**) Overview of the experiment. Six induced pluripotent stem cell (iPSC) lines from four donors were differentiated to cortical neural fate and assayed at days 20, 40, and 70 of the differentiation using CP and scRNA-seq. (**b**) UMAP of the CP dataset at the single-cell level. Cells are colored and labeled by unsupervised Leiden clustering. A composite image representative of cells belonging to each CP cluster is indicated by arrows (staining legend bottom-right). (**c**) Heatmap of CP feature-module activation

*Figure 1 continued on next page*

*Figure 1 continued*

scores. 541 cells (minimum cluster size) were drawn from each CP cluster to ensure an equal contribution of each CP cluster to the heatmap. The CP feature modules represent sets of CP features that are highly correlated. The right-most column contains selected highly enriched terms from GSEA of CP feature modules. (**d**) Differentiation day (time points) projected on the CP UMAP.

The online version of this article includes the following figure supplement(s) for figure 1:

**Figure supplement 1.** Benchmarking of *in vitro* cell types.

**Figure supplement 2.** Cell Painting (CP) analysis workflow.

**Figure supplement 3.** Additional Cell Painting (CP) results.

## Results

### CP at the single-cell level captures developmental progression

Cellular organelles such as mitochondria and the ER play important roles in neurodevelopment, specifically in meeting dynamic bioenergetic and metabolic requirements (*Zhang et al., 2022*). To characterize the cellular and morphological features of developing neurons *in vitro*, we differentiated iPSCs from four donors toward cortical neuronal fate and profiled them using CP at days 20, 40, and 70 of the differentiation (*Figure 1a*, *Figure 1—figure supplement 1*, *Table 1*, *Supplementary file 1a–d*, Methods). We obtained image-based features such as fluorescence intensities, texture, and cell shape measures from each CP channel, which together represent the overall morphological profile of the cell. These features were extracted into a matrix per single cell and, after quality control and regularization (Methods, *Figure 1—figure supplement 2*), we obtained a feature matrix of 223 features from 54,415 cells. Based on observations from performing well-to-well correlation, we found that differentiation time point was the major source of variation (*Figure 1—figure supplement 3a*).

To maintain and explore heterogeneity in the dataset, the feature matrix was then analyzed in an analogous approach to scRNA-seq data. Leiden clusters projected on a Uniform Manifold Approximation and Projection (UMAP) (*Figure 1b*) were used to explore the main phenotypic states captured by CP. We found that the unsupervised clusters captured variation in both cell size and organelle staining intensities. In light of the high modularity of CP features, we defined feature modules based on the correlation between CP features (*Figure 1c*, *Figure 1—figure supplement 3b*). Correlated features within the same module usually corresponded to multiple aspects of the same measurement, for example, cell area and perimeter, or the average and minimum intensity of each channel. Interestingly, there was a global correlation between the channel comprising the cell membrane, Golgi apparatus, actin cytoskeleton and nucleoli (CGAN), and the ER channel (*Figure 1—figure supplement 3b*). In contrast, some pairs of CP features were highly anticorrelated. In our study, this was observed between cell size and density, suggesting that smaller cells are more likely to be found in areas of higher density (*Figure 1—figure supplement 3c*). Furthermore, a near-perfect negative correlation was detected between texture values related to the uniformity of the channel (angular second moment) and the intensity within each channel. This phenomenon suggests that these two measurements may represent inverse aspects of the same underlying characteristic.

The distribution of feature values shows that despite the difference between time points, CP was able to capture a relative continuity across them. Clusters composed mainly of day 20 cells were characterized by high intensities for the mitochondria channel (*Figure 1c, d*), and clusters with mainly day 40 cells by medium mitochondria intensity and high ER intensity. Interestingly, day 70 was the most heterogeneous time point with three distinct profiles: cells with a large area that were close to d40 cells in terms of features value (d70.bigCells), cells with very low mitochondria intensity (d70.mitoNeg), and cells that clustered with d20 cells (d70_20.mito).

### Biological annotation of image features reveals separation of broad neuronal cell types

To enable the interpretation of the cellular and morphological variability captured by CP clusters, we linked the feature space with gene expression. For this, we additionally profiled the transcriptomes of developing neurons from the same four donors at days 20, 40, and 70 of the differentiation using scRNA-seq (*Figure 2a*). We then modeled the relationship between CP and scRNA-seq similarly to *Haghighi et al., 2022*, by using the common experimental design between the two assays to

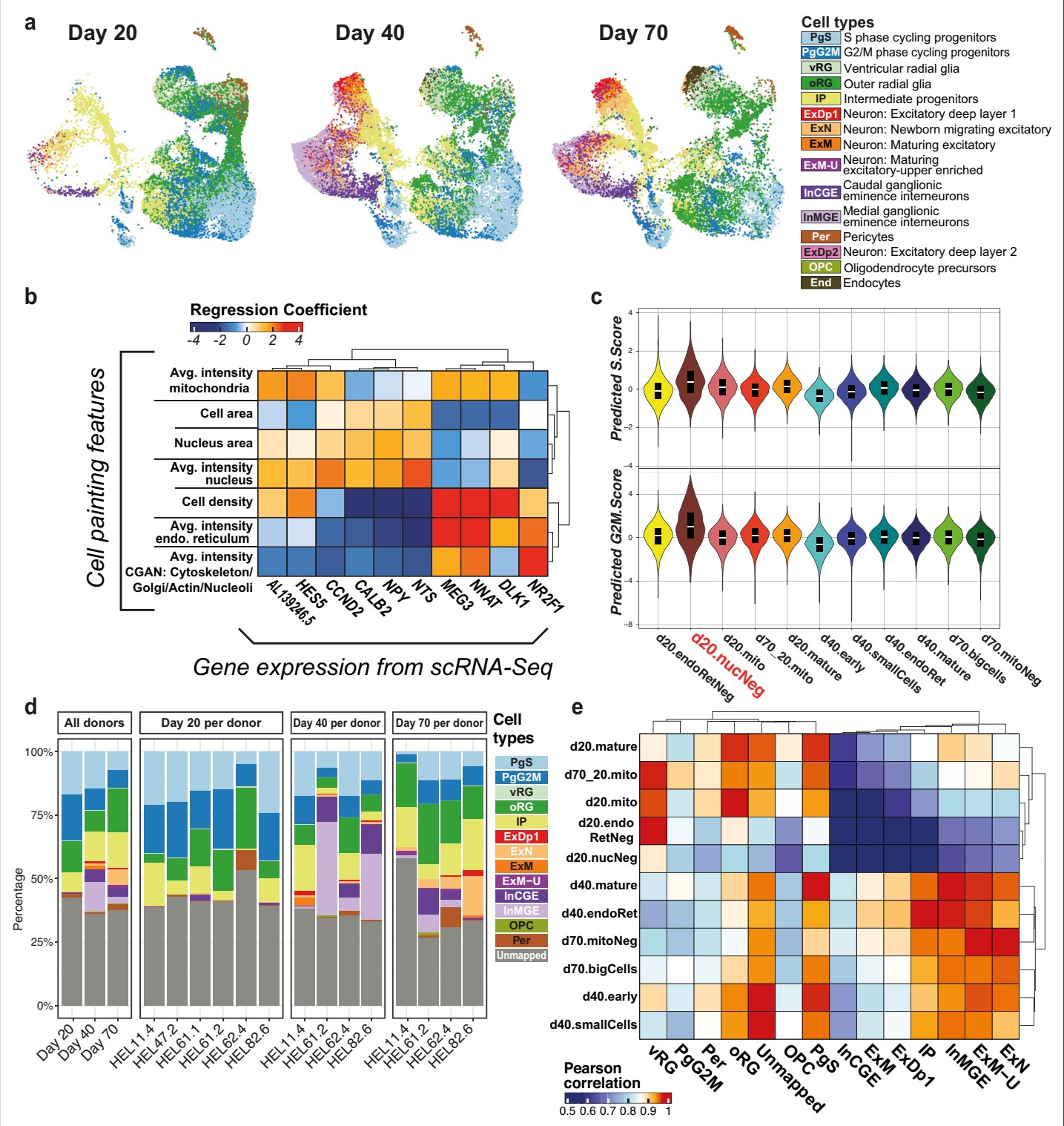

**Figure 2.** Joint profiling of neural cell types observed *in vitro*. (**a**) UMAP of cell types as identified by scRNA-seq analysis, split by time point. (**b**) Representative heatmap of the model linking expression of marker genes (columns) to Cell Painting (CP) image-based features (rows). (**c**) Violin plots of predicted cell cycle scores per CP cluster. (**d**) Cell type composition from the scRNA-seq dataset. The facet on the left represents aggregated cell type proportions across donors per time point, while the other facets represent individual time points, depicting cell type proportions for each donor. (**e**) Correlation heatmap of cell types to CP clusters, between average of CP features values per cell cluster and average of predicted CP features from scRNA-seq. Abbreviations: PgS – S phase progenitors, PgG2M – G2M phase progenitors, vRG – ventral radial glia, oRG – outer radial glia, IP – intermediate progenitors, ExDp1/2 – excitatory deep layer neurons 1/2, ExN – migrating excitatory neurons, ExM – maturing excitatory neurons,

*Figure 2 continued on next page*

*Figure 2 continued*

ExM-U – upper-layer enriched maturing excitatory neurons, InCGE – caudal ganglionic eminence interneurons, InMGE – medial ganglionic eminence interneurons, OPC – oligodendrocyte precursors, Per – pericytes, End – endothelial cells.

The online version of this article includes the following figure supplement(s) for figure 2:

**Figure supplement 1.** Multi-modal analysis workflow.

**Figure supplement 2.** Cell type annotations in scRNA-seq.

**Figure supplement 3.** *In vitro* to fetal cell type correlations.

train lasso regression models used to predict the corresponding CP feature values of the scRNA-seq dataset (*Figure 2b*, *Figure 2—figure supplement 1*, Methods). We used the coefficients matrix from the resulting model to perform a functional enrichment of previously identified CP feature modules, using the median of gene coefficients per feature module as an input to Gene Set Enrichment Analysis (GSEA) (*Figure 1c*; *Mootha et al., 2003*; *Subramanian et al., 2005*). The analysis revealed a predominant focus on the proteasome, extracellular matrix, and mitochondrial respiration. Despite redundancy among the enriched terms observed, this pattern underscores the property of CP to predominantly capture variation related to cell morphology rather than specific pathway regulation. In most cases, the enriched terms per module were closely linked to the module-associated features, increasing the confidence in our model. For example, in a module composed of features associated with the ER channel, the most enriched terms were related to the ER membrane. Interestingly, other general terms such as intensity of mitochondria were linked to mitochondrial ribosomes rather than mitochondrial respiration or oxidative phosphorylation.

A small pool of d20 cells (d20.nucNeg) was particularly linked with mitosis. To further explore the possible link between CP and cell cycle, we predicted the cell cycle scores (G2M score, S score) in the CP dataset by using the predicted CP features matrix on scRNA-seq profiled cells and their annotated cell cycle scores (Methods). Using our regression model, 'd20.nucNeg' cells showed the highest median values for both G2M score and S score (*Figure 2c*), and, coupled with their particular morphology, suggest that CP is able to capture cycling cells at the progenitor stage.

Neuronal differentiation is a tightly regulated process, generating heterogeneous cell types in a temporal manner. To resolve cell types across the three time points of the CP dataset, we first annotated 62,929 cells present in our scRNA-seq dataset. It has been established previously that *in vitro* generated neurons more closely resemble fetal rather than adult neuronal cell types (*Handel et al., 2016*; *van de Leemput et al., 2014*). We thus used the reference mapping approach from the Seurat package (*Hao et al., 2021*) to transfer cell type labels from a well-annotated mid-gestation (gestational

**Table 1.** Cell lines used in the study with their corresponding donor information, derivation method, registration (https://hpscreg.eu/), and karyotype cytogenetic nomenclature.

| Name | Cell type | Diagnosis | Sex | Derivation method | Characterized as iPSC | Marker expression | Ref. | Registry info | Karyotype |
|---|---|---|---|---|---|---|---|---|---|
| HEL11.4 | Skin fibroblast | Unaffected | Male | Retrovirus | Yes | Yes | *Mikkola et al., 2013* | UHi007-B | 46,X,inv(Y)(p11q11),add(1)(q12q21) |
| HEL47.2 | Skin fibroblast | Unaffected | Male | Sendai virus | Yes | Yes | *Trokovic et al., 2015* | UHi007-A | 46,X,inv(Y)(p11q11) |
| HEL62.4 | Skin fibroblast | Unaffected | Female | Sendai virus | Yes | Yes | | UHi020-A | 47,XX,+12[2]/46,XX[18] |
| HEL61.1 | Skin fibroblast | Unaffected | Female | Sendai virus | Yes | Yes | | UHi021-A | 46, XX |
| HEL61.2 | Skin fibroblast | Unaffected | Female | Sendai virus | Yes | Yes | | UHi021-B | 46, XX |
| HEL82.6 | Skin fibroblast | Unaffected | Female | Sendai virus | Yes | Yes | | UHi022-A | 46, XX |
| aask4 | Skin fibroblast | Kabuki syndrome | Female | Sendai virus | Yes | Yes | | WTSIi580-A | 46, XX |
| hoik1 | Skin fibroblast | Unaffected | Female | Sendai virus | Yes | Yes | | WTSIi026-A | 46,XX,t(6;13)(p11;q11) |
| ierp4 | Skin fibroblast | Kabuki syndrome | Female | Sendai virus | Yes | Yes | | WTSIi469-A | 47,XXX,t(2;5)(q12;q35.1)[9]/46,XX,t(2;5)(q12;q35.1)[11] |
| oadp4 | Skin fibroblast | Kabuki syndrome | Male | Sendai virus | Yes | Yes | | WTSIi558-A | 46, XY |
| oikd4 | Skin fibroblast | Unaffected | Female | Sendai virus | Yes | Yes | | WTSIi260-A | 46, XX |
| qeti2 | Skin fibroblast | Kabuki syndrome | Male | Sendai virus | Yes | Yes | | WTSIi526-A | 46, XY |

weeks 17–18) fetal reference (*Polioudakis et al., 2019*) onto our query dataset (*Figure 2—figure supplement 2*), setting a mapping quality threshold of 0.5 (Methods, *Figure 2—figure supplement 3*). With this approach, we annotated 60% of our cells with high confidence and without being limited to a few canonical markers, aiming to better describe the dynamics of *in vitro* cortical differentiation. We identified 14 cell types produced across the three time points, capturing a large portion of the heterogeneity seen in developing neurons (*Figure 2a and d*). These include various progenitor cell types (cycling progenitors, ventral and outer radial glia [oRG]) as well as IPs and maturing neuronal cells (deep layer and maturing upper-layer excitatory neurons). We also detected inhibitory neurons within our cortical system (MGE- and CGE-like interneurons), as has been reported recently (*Delgado et al., 2022*).

We used the predicted CP feature matrix to assess the correlation between CP clusters and the cell types inferred from scRNA-seq (*Figure 2e*, Methods). We found that CP clusters captured the two largest cell classes – neural progenitors and neuronal cells, with each CP cluster correlating to multiple cell types within these classes. Individual cell types could not be distinguished based on CP clusters; however, this is unsurprising given the highly generalizable nature of the assay. This demonstrates the efficacy of CP in classifying broad categories of cells yet highlights that the variability it captures in morphology falls outside the current scope of cell types defined in the transcriptomic space.

## Neurodevelopment *in vitro* follows known trajectories and recapitulates cell type heterogeneity seen *in vivo*

Neurodevelopment *in vivo* is characterized by a tightly regulated developmental trajectory, from neural progenitors to radial glia and, via IPs, to maturing neurons. Given the limitations in cell type resolution using CP, we sought to instead assess the morphological changes along this trajectory. Using the scRNA-seq data, we first estimated the pseudotime of our differentiating cells across the three time points (Methods) and found that the developing cells *in vitro* follow a similar trajectory as seen *in vivo* (*Figure 3a*). Additionally, by computing modules of co-expressed genes changing as a function of pseudotime with the Monocle3 toolkit (*Trapnell et al., 2014*), we identified gene sets linked to the development of individual cell types (*Figure 3—figure supplement 1a*). These modules were enriched for many expected biological processes, such as 'nuclear division' in PgG2M cells and 'ribonucleoprotein complex biogenesis' in PgS (*Figure 3—figure supplement 1b–f*). The IP-associated module was tagged by terms such as 'cell fate commitment' and 'channel activity', indicative of the transitory role these cells play between progenitors and neurons. Neuronal modules were characterized by formation and regulation of synapses as well as ion channel activity, with the inhibitory neuron-associated module linked to GABAergic signaling.

From the CP data, we observed a distinct pattern of mitochondrial features across the UMAP, with mitochondrial intensity-related features more prominently associated with day 20 cells and decreasing toward day 70, whereas mitochondrial uniformity features ('angular second moment') followed the opposing trend (*Figure 3b*). To associate this trend with the developmental trajectory, we projected these CP features onto the scRNA-seq UMAP confirming that the trend is maintained (*Figure 3c*); mitochondrial intensity is higher amongst progenitor cell types, whereas uniformity is higher amongst mature/maturing neurons, representing uniform (low intensity) texture. The notable exception is the region of the UMAP occupied by inhibitory neurons, specifically MGE-like interneurons, which instead have high mitochondrial intensity and low uniformity, likely due to their more immature (and thus morphologically progenitor-like) state.

Given the cell-type specificity of mitochondrial CP features, we sought to determine whether we could detect changes in the transcription profile linked to mitochondrial metabolism. It has been reported previously that NPCs meet their energetic requirements primarily through glycolytic pathways, and the switch to neuronal fate is associated with a switch to dependency on oxidative phosphorylation (OxPhos) (*Romero-Morales and Gama, 2022*). Gene set variation analysis (GSVA) on pseudobulked cell types for both glycolysis and OxPhos (Methods) confirmed that most neuronal cell types had lower glycolytic dependency, with all excitatory cell types showing maximum activation for OxPhos, except for a subset of excitatory deep layer neurons (ExDp1) (*Figure 3d*). We further tested for enrichment of gene ontology (GO) terms associated with mitochondria in the gene expression patterns per cell type, finding a clear link between the progenitor cell types and mitochondrial ribosomal subunits, whereas excitatory neuronal cell types were linked to mitochondrial

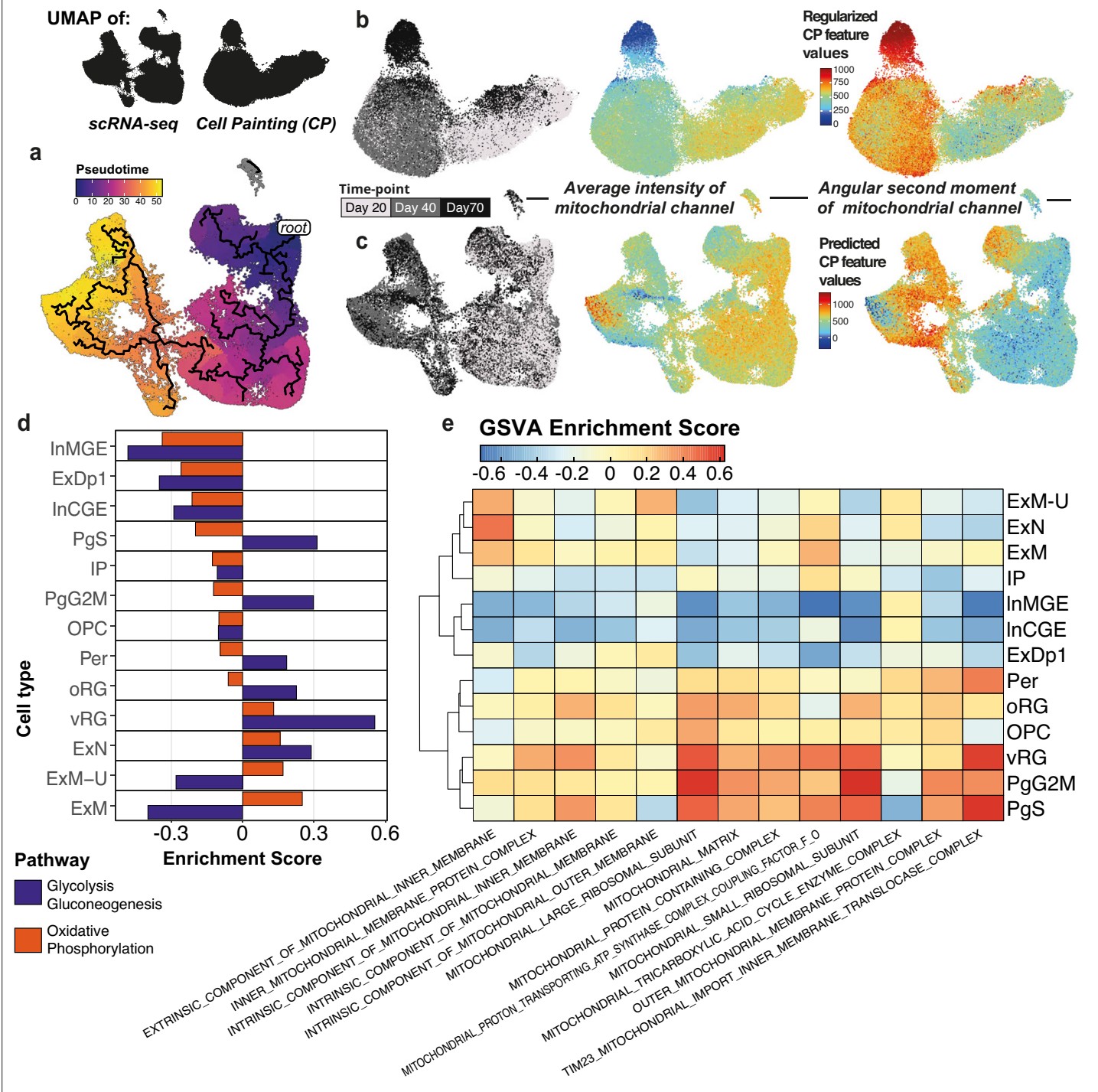

**Figure 3.** Mitochondrial feature dynamics in Cell Painting (CP) and gene expression. (**a**) Predicted pseudotime trajectory across time points in scRNA-seq showing progression from radial glia and progenitors, via intermediate progenitors, toward maturing excitatory and inhibitory neurons. Pseudotime is rooted in the predicted earliest node within day 20. (**b**) CP UMAPs with differentiation day (left), feature values for mitochondrial intensity (middle), and mitochondrial texture (right) projected. (**c**) scRNA-seq UMAPs with differentiation day (left), predicted values for the CP features mitochondrial intensity (middle) and texture (right) projected. (**d**) Gene set variation analysis (GSVA) enrichment scores for glycolysis and oxidative phosphorylation pathways per pseudobulked cell type. Enrichment scores indicate a higher activation score of OxPhos in excitatory neurons, except for the deep layer 1 subtype (ExDp1). (**e**) GSVA enrichment scores for mitochondria-associated gene ontology (GO) terms per pseudobulked cell type.

The online version of this article includes the following figure supplement(s) for figure 3:

**Figure supplement 1.** Functional enrichment across the developmental trajectory.

membrane components, known to be linked to OxPhos (*Figure 3e*). These findings complement the trend observed from the functional enrichment of CP feature modules, where day 20 clusters were enriched in mitochondrial ribosome and mitosis, day 40 in transcriptional regulation, and day 70 in oxidative phosphorylation (*Figure 1c*). Overall, this emphasizes the ability of the morphological profiling to capture phenotypes relevant to those from transcriptomics.

## Donor-specific effects drive neuronal cell fate determination and are captured by both modalities

Inhibitory GABAergic interneurons play a key role in cortical circuits by balancing the excitatory/inhibitory activity and by regulating the formation of synapses. Abnormalities in interneuron development, function, or migration have been associated with autism spectrum disorders (*Contractor et al., 2021*; *Paulsen et al., 2022*) and other NDDs (*Ramamoorthi and Lin, 2011*; *Meng et al., 2023*). Of the neuronal cells identified in our dataset (21% of all cells), nearly 40% were annotated as inhibitory interneurons of either caudal or medial ganglionic eminence (InCGE/InMGE).

Although interneurons have typically been described to originate from brain regions outside of the cortex, *in vitro* studies have previously reported the generation of GABAergic neurons from cortical progenitors (*Alzu'bi et al., 2017*; *Strano et al., 2020*). A recent study from *Delgado et al., 2022* confirmed via clonal lineage tracing studies that a subpopulation of cortically born GABAergic neurons was transcriptionally similar to ventrally derived cortical interneurons, but instead arose from cortical progenitors. The InCGE cells from our dataset were characterized by expression of the *DLX* family of genes, as well as *GAD2*, *DCX*, and *MEIS2*. InMGE cells expressed a subset of canonical markers such as *SST*, *DCX*, and *MEIS2* but, similar to Delgado *et al.*, differed from true ganglionic eminence interneurons in lack of expression of *LHX6* and *NKX2.1* (*Figure 4—figure supplement 1a*). The production of interneurons in our system is valuable to better recapitulate human fetal development considering the relevance of the inhibitory component in neural circuits.

The proximal clustering of these interneurons with excitatory neuronal cell types prompted us to explore the pathways that differed between them. Previous studies both in mice (*Zhang et al., 2020*) and *in vitro* cultures (*Strano et al., 2020*) have indicated the interplay of WNT and SHH signaling as a key factor in maintaining the excitatory/inhibitory balance. However, GSVA analysis of KEGG genes associated with these pathways did not show a consistent upregulation of WNT signaling in all excitatory cell types, nor vice versa with Hedgehog signaling for inhibitory cell types (*Figure 4—figure supplement 1b, c*). Instead, we found lower values of enrichment scores for oxidative phosphorylation pathway-associated genes in inhibitory than in excitatory neurons (*Figure 3d*, also seen from GSEA, *Figure 4—figure supplement 1d*).

Additionally, as noted earlier, MGE-like interneurons showed marked differences from the other neuronal cell types in CP-based mitochondrial features (*Figure 3c*). A single donor (HEL61.2) presented an excess of this inhibitory neuronal subtype across time points (*Figure 4a*). This same donor was overrepresented in the CP cluster 'd20.endoRetNeg', characterized by lowered intensity of the ER channel, and underrepresented in the ER-intense 'd40.endoRet' cluster (*Figure 4b*), pointing toward donor-specific ER effects potentially linked to inhibitory cell fate. Interestingly, we also observed an effect of lowered ribosome-associated genes in gene expression data – the two donors (HEL61.2 and HEL82.6) accounting for 80% of inhibitory cell types at days 40 and 70 showed an overall decrease in the KEGG ribosome pathway at these time points (*Figure 4c*). Complementarily, a GSVA of mitochondria-associated GO terms across our in vitro cell types (*Figure 3e*) revealed an overall depletion of mitochondrial ribosome-associated genes in inhibitory neurons compared to other cell types, as has been previously reported (*Wynne et al., 2021*). To assess how generalizable our findings from this individual donor were, we computed GSVA scores for KEGG pathways in five additional cell lines across multiple time points from published organoid datasets (*Bhaduri et al., 2020*; *He et al., 2024*). We confirmed an overall negative trend between inhibitory neuron proportions and ribosomal activity (*Figure 4—figure supplement 1e*), showcasing the ability of CP to capture relevant donor-specific variation.

In order to elucidate which genes are responsible for the bifurcation between the excitatory and inhibitory fate in our *in vitro* differentiation, we focused on the branch point along the pseudotime trajectory that either produces inhibitory interneurons or IPs, which in turn give rise to excitatory neurons. Within this branch point, we identified modules of genes that were differentially expressed

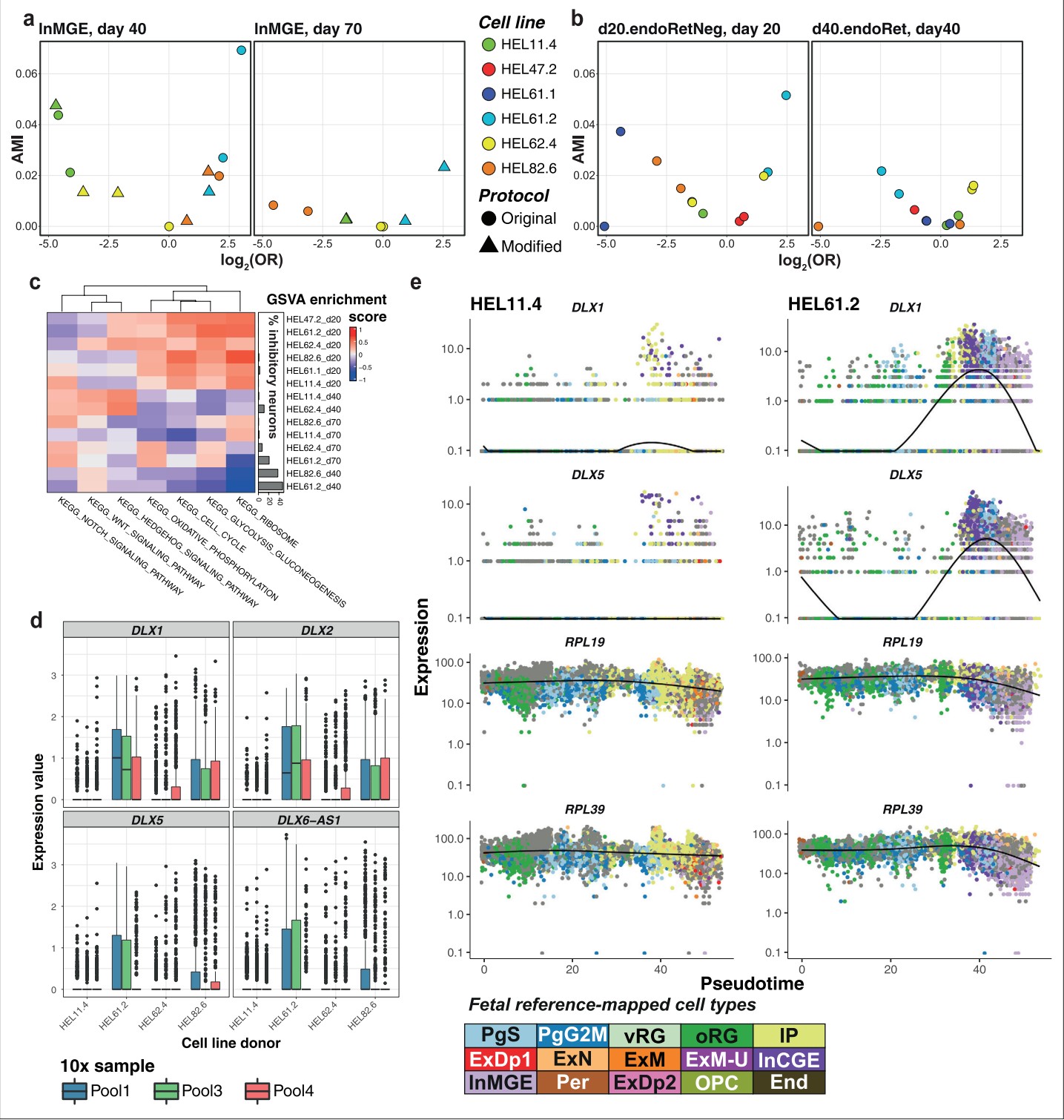

**Figure 4.** Mechanisms of donor-specific inhibitory neuron production. (**a, b**) Association plots between the cell lines used in this study and cell type/cluster annotations. x-axis: log₂ of the odds ratio, y-axis: AMI (Adjusted Mutual Information): AMI score here measures dependence between proportions of cell types/clusters present and cell line of origin, with an AMI score closer to 1 indicating a high degree of agreement, while values near 0 suggest little to no similarity beyond random chance. (**a**) Association of cell line abundance with InMGE cells from scRNA-seq at days 40 and 70. Each point represents aggregate expression per technical replicate, donor, and the differentiation protocol used. (**b**) Association of cell line abundance with the 'd20.endoRetNeg' cluster at d20 and 'd40.endoRet' at day 40, from Cell Painting (CP) data. Each point represents aggregate expression of all wells per induction replicate and donor. (**c**) Gene set variation analysis (GSVA) enrichment scores from curated KEGG pathways between experimental

*Figure 4 continued on next page*

Figure 4 continued

populations (donor x time point) with hierarchical clustering on both axes. Annotation bar plot to the right of the heatmap shows the percentage of cells produced by each experimental population annotated as inhibitory cell types (InCGE, InMGE, and InhN-O). (**d**) Relative expression per donor (log-normalized values) of inhibitory interneuron-associated transcription factors within the progenitor pool (oRG + vRG + PgG2M + PgS) at day 40. The expression per donor is grouped per 10x sample. (**e**) Expression of interneuron-associated *DLX* genes and ribosomal subunits *RPL39* and *RPL19,* along pseudotime in two donors shown to produce a high (HEL61.2, left) and a low (HEL11.4, right) proportion of inhibitory neurons, respectively. Point colors represent fetal-annotated cell types.

The online version of this article includes the following figure supplement(s) for figure 4:

**Figure supplement 1.** Characterization of inhibitory neurons produced *in vitro*.

as a function of the pseudotime (Methods). As expected, the module marking the inhibitory branch (module 12, *Figure 4—figure supplement 1f*) consisted of TFs known to drive interneuron fate such as the *DLX* family of genes, which was overexpressed in the donor HEL61.2 in day 40 progenitors (*Figure 4d*). In donors producing inhibitory neurons, expression of the *DLX* family increased with pseudotime, peaking at interneuron production. Concordantly, the expression of genes driving the differential activation of the KEGG ribosome pathway (*RPL39* and *RPL19*) also decreased in donors producing inhibitory neurons (*Figure 4e*). Altogether, our data is indicative of lowered ribosomal activity and mitochondrial differences between excitatory and inhibitory neurons overall.

## Metabolic differences shape high- versus low-quality cells produced *in vitro*

It is well established that *in vitro* cell types do not completely resemble their *in vivo* counterparts. When mapped to the reference dataset, 60% of our cells mapped at high confidence to fetal cell types, while the remaining 40% remained 'unmapped'. These unmapped cells were found to be present across the UMAP (*Figure 5a*), likely indicating that cells did not fully resemble the fetal transcriptomes and/or were transitioning between cell types, rather than being a single missing or unannotated cell type. By assigning the highest scoring cell type label to each cell in the unmapped fraction of cells, we divided our dataset into high- and low-quality (HQ/LQ) cells for each annotated cell type (Methods). The LQ cells still expressed canonical markers of the cell type they were closest to, although at lower levels, except in the case of oRG and IPs (*Figure 5—figure supplement 1a*). Additionally, LQ cells did not show consistent differences in pseudotime scores as compared to HQ cells of the same cell type (*Figure 5—figure supplement 1b*), which would have been indicative of transitioning cell states.

Given this, we assessed what transcriptomic features were driving their lower mapping scores. By comparing differential activation of developmentally relevant pathways between high- and low-quality cells per cell type, we identified those that differed (*Figure 5b*). Importantly, although metabolic processes seemed to be implicated overall, we did not see consistent changes across all cell types, rather observing specific changes driving the LQ version of each cell type, such as DNA replication and glycolysis in progenitor cell types (vRG, oRG, and IP). Also, we observed the highest steroid biosynthesis activation in HQ maturing excitatory neurons (ExM) compared to LQ, but the opposite trend was observed within progenitors (PgS and PgG2M).

Aberrant cellular metabolism has previously been reported in other *in vitro* systems, often highlighted in the oxidative phosphorylation and glycolytic pathways (*He et al., 2024*; *Uzquiano et al., 2022*). In cortical organoids, it has been seen that a fraction of cells does not recapitulate distinct fetal cellular identities (*Bhaduri et al., 2020*). Hypothesizing that our unmapped cells represented a similar fraction, we annotated our unmapped cells to cortical organoid cell types from *Bhaduri et al., 2020* using the same label-transfer method and threshold described above (Methods). We found that more than 50% of the previously unmapped cells could be attributed to 'pan-radial glial' or 'pan-neuronal' cell types across all three time points (*Figure 5c*). These were described in the original publication as broad progenitor or neuronal cell classes that did not express features distinctive to any specific fetal subtype.

Next, we compared the differentially activated pathways between cells of the closest cell types that passed this bottleneck of subtype acquisition to those that maintained a pan-cell identity by identifying modules of the top genes driving differentially activated pathways in these cell types and scoring these gene modules per single cell (Methods). Similar to differences observed between high- and low-quality progenitor cell types, pan-radial glia differed in ribosome-associated pathways. Pan-neuronal

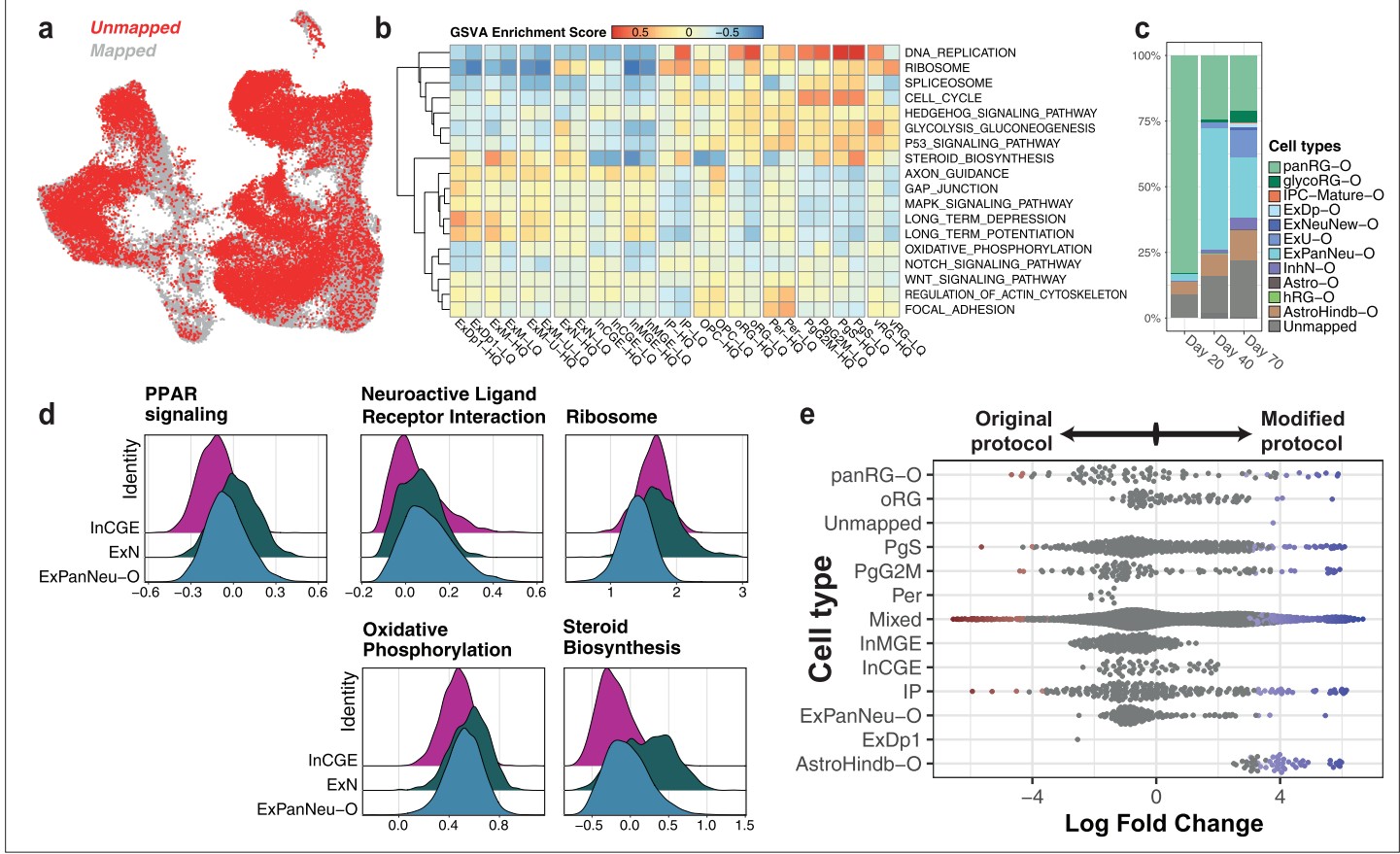

**Figure 5.** Characterization of low-quality cells within the scRNA-seq dataset. (**a**) UMAP of scRNA-seq data across time points with unmapped cells highlighted in red. (**b**) Enrichment scores from gene set variation analysis (GSVA, across cell types) of key pathways driving the differences between high- and low-quality subsets per pseudobulked cell type. (**c**) Percentage of cells from the initial unmapped fraction that align to organoid cell types (Bhaduri et al.) per time point. (**d**) Ridge plots representing the distribution of module activation scores of key differentially activated pathways between pan-neuronal (ExPanNeu-O), excitatory (ExN), and inhibitory (InCGE) neuronal cell types. (**e**) Differential abundance of cell types produced by the original (left) or the modified (right) versions of the differentiation protocol, computed by miloR. Abbreviations: panRG – pan radial glia, glycoRG – glycolytic radial glia, IPC-Mature – mature intermediate progenitors, ExDp – excitatory deep layer, ExNeuNew – newborn excitatory neurons, ExU – excitatory upper layer, ExPanNeu – pan-neuronal (excitatory), InhN – inhibitory neurons, Astro – astrocytes, hRG – hindbrain radial glia, AstroHindb – hindbrain astrocytes. The suffix '-O' differentiates organoid cell type annotations from their fetal counterparts.

The online version of this article includes the following figure supplement(s) for figure 5:

**Figure supplement 1.** LQ and HQ cell type subsets show similarities in expression and pseudotime.

**Figure supplement 2.** scRNA-seq quality control.

**Figure supplement 3.** Reproducibility of replicates in scRNA-seq: HEL11.4.

**Figure supplement 4.** Reproducibility of replicates in scRNA-seq: HEL47.2.

**Figure supplement 5.** Reproducibility of replicates in scRNA-seq: HEL61.1.

**Figure supplement 6.** Reproducibility of replicates in scRNA-seq: HEL61.2.

**Figure supplement 7.** Reproducibility of replicates in scRNA-seq: HEL62.4.

**Figure supplement 8.** Reproducibility of replicates in scRNA-seq: HEL82.6.

cells (ExPanNeu-O), however, were characterized by features of both excitatory and inhibitory neurons, differing again metabolically such as in oxidative phosphorylation or steroid biosynthesis (*Figure 5d*). We found that overall, pan-neuronal cells appear to be intermediate to either excitatory or inhibitory neuron specification, except for the ribosome pathway.

With the inclusion of the second step of annotation, we identified cell types that were initially missed, mainly astrocytes (*Figure 5c*). While expected to be produced in this protocol, they were absent in the fetal reference, and thus, the first annotation step alone did not identify them. Of note,

our experimental workflow involved the use of two versions of the differentiation protocol (original/modified, see Methods), differing in enzyme usage for cell dissociation (*Figure 5—figure supplements 2–8* see *Supplementary file 1b* for details). To check for potential effects of the protocol on the generation of cell types, we used *miloR* (*Dann et al., 2022*) to test for differential abundance (DA) of annotated cell types between the protocol versions. We found that the astrocyte population (AstroHindb-O) was overrepresented in the modified version of the protocol at day 40 (*Figure 5e*).

### *In vitro*-derived cells capture disease-relevant cellular phenotypes

Finally, to evaluate the relevance of our *in vitro* cortical neurons for disease modeling, we used stratified LD score regression (*Finucane et al., 2018*) to quantify how much heritability of common diseases and other complex traits is enriched within genes that are markers of the cell types generated *in vitro*. We analyzed GWAS summary statistics from a set of 79 common traits including 12 brain-related phenotypes (*Supplementary file 1g*, Methods). There was a clear enrichment of significant brain-related associations when accounting for all tests (Fisher test, $p = 9.62 \times 10^{-13}$). Comparison of traits captured by *in vitro* cell types (*Figure 6a*, *Figure 6—figure supplement 1a*) to those captured by GTEx brain tissues (*GTEx Consortium, 2013*; *Figure 6—figure supplement 1b*) shows that iPSC-derived neurons are relevant for multiple brain-related traits that are associated with the cortex/frontal cortex. We captured certain traits missed in GTEx tissue such as risk tolerance (Linner2019risk) in deep layer excitatory and MGE-like inhibitory neurons, as predicted by *Karlsson Linnér et al., 2019*. Similarly, we found a strong association between depression (Depression2018) and excitatory deep layer neurons, as well as maturing excitatory neurons (*Nagy et al., 2020*). We also observed a high enrichment of educational attainment-associated genes in InMGE neurons (EduYear), concordant with reports of the role of inhibitory neurons in learning and memory (*Ramamoorthi and Lin, 2011*).

In addition to brain-related complex traits, iPSC-derived neuronal models are widely used to model rare NDDs, given their ability to recapitulate cell lineages from very early developmental time points and their transcriptional similarity to fetal, rather than adult cell types. To identify which of our cell types expressed genes causally linked with NDDs, we evaluated the cell-type-specific expression of brain-specific developmental disorder genes from the Deciphering Developmental Disorders (DDD) study (*Wright et al., 2015*). Interestingly, we observed that while approximately 45% of all high-confidence, brain-related DDD genes ($n$ = 719) exhibit their highest expression levels in mature neuronal cell types, progenitor cell types also captured another ~30% of genes associated with developmental delay (*Figure 6—figure supplement 1c*). Focusing on a subset of DDD genes ($n$ = 72) known for their high intolerance to loss-of-function mutations (Methods), we found them to be enriched across the gene markers of neuronal cell types, as well as in progenitors (IP and PgG2M) (*Figure 6a*, upper).

We next sought to delineate whether CP features could also capture disease-relevant traits. We performed stratified LD score regression analysis on the genes predictive of the most representative features per CP feature module (Methods), using the same set of brain-related traits. We found that while CP feature modules did not correspond to individual cell types identified from scRNA-seq (*Figure 2e*), modules related to nuclear shape (misc – Nuc_shape_eccentricity) and intensity of cytoskeletal structure (integrInt.cytoStruct) nonetheless captured heritability for traits such as schizophrenia and ADHD (*Figure 6b*). The same feature modules along with those associated with mitochondrial texture (angMoment.mito) and nuclear entropy (entropy.nuc) were also enriched for the DDD brain signature ($n$ = 72 genes) (*Figure 6b*, upper).

Finally, to directly test if CP could identify disease-relevant phenotypes, we performed an independent differentiation of iPSC lines from six donors including four individuals with Kabuki syndrome (KS), a rare NDD, to neural progenitors. We profiled the cells with CP at day 20, integrated the new CP dataset with the original, and transferred the original cluster labels (*Figure 1b*) to the integrated data using a nearest-neighbors approach (Methods, *Figure 6—figure supplement 2*). DA analysis between KS and healthy donor samples, stratified by the transferred cluster labels, revealed clusters that were significantly enriched or depleted in KS samples. Specifically, the clusters 'd20.endoRetNeg' and 'd20.nucNeg' were differentially abundant in healthy donor lines across both batches (*Figure 6c*); notably, 'd20.nucNeg' was previously shown to be associated with increased cell cycle activity (*Figures 1c and 2c*). On the other hand, the clusters upregulated in KS lines ('d40.early' and 'd40.smallCells') represented d40-like cell states, despite being derived from day 20 differentiations (*Figure 6c*). These findings point to a precocious differentiation phenotype and decreased cell cycling, previously observed

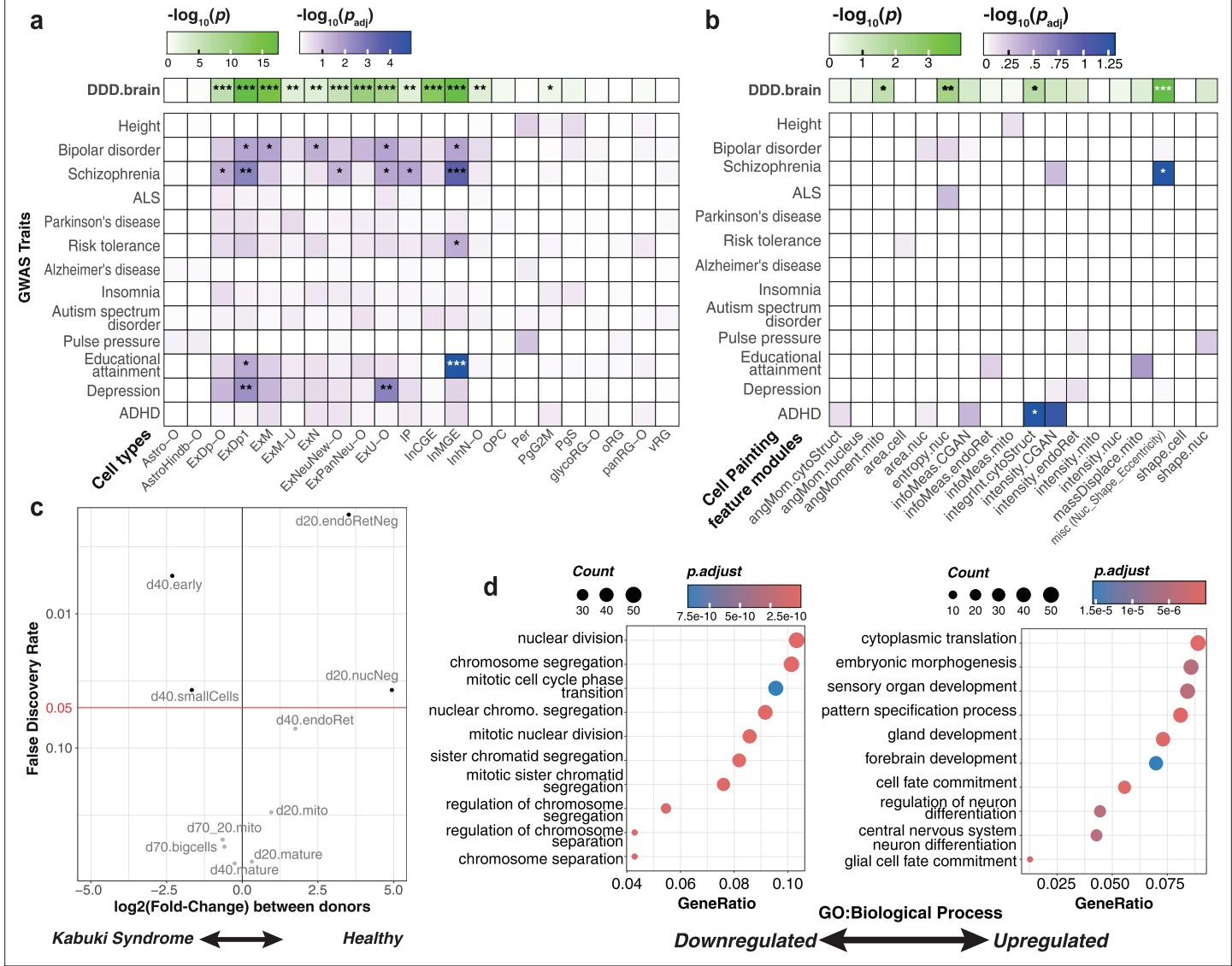

**Figure 6.** *In vitro* neurons capture brain-relevant traits. (**a**) Stratified linkage disequilibrium (LD) score regression analysis is shown for selected brain-related traits per cell type from our *in vitro* differentiation. Tile color represents the corresponding p-values after multiple testing correction across traits and cell types, with the significance level indicated as follows: *pAdj < 0.05, **pAdj < 0.01, and ***pAdj < 0.001. The top-most row represents enrichment across brain-specific genes associated with developmental disorders (n=68), with tile color representing p-values from a one-tailed Fisher's exact test (*p < 0.05, **p < 0.01, and ***p < 0.001). (**b**) Stratified LD score regression analysis and Deciphering Developmental Disorders (DDD)-brain enrichment as in (**a**) per Cell Painting (CP) feature module. (**c**) Differential abundance of CP clusters between d20 samples from eight healthy individuals and four individuals with Kabuki syndrome (KS). (**d**) Visualization of gene ontology (GO): Biological Process terms enriched within the top genes predictive of CP clusters which are differentially abundant in either healthy donors (left, downregulated) or KS donors (right, upregulated).

The online version of this article includes the following figure supplement(s) for figure 6:

**Figure supplement 1.** Cell-type specificity of brain- and non-brain-related traits.

**Figure supplement 2.** Integration of the two Cell Painting (CP) batches.

**Figure supplement 3.** Gene ontology (GO) enrichment of differentially abundant clusters between Kabuki syndrome (KS) samples and healthy donors.

in KS and other NDDs (*Carosso et al., 2019*; *Jhanji et al., 2024*) demonstrating the utility of CP-derived morphological features in revealing disease phenotypes.

To further contextualize these findings, we applied results from our existing linear regression model (Methods, *Figure 2b*) to identify the biological processes enriched within the differentially abundant clusters. Overrepresentation analysis (ORA) of top predictive genes for these clusters highlighted a downregulation of cell cycle-related pathways and an upregulation of patterning and neuronal

differentiation in KS versus healthy donor lines at day 20 (*Figure 6d*). The top features marking the clusters depleted in KS included ER texture metrics ('InvDiffMnt' and 'AngScndMoment'), as well as nuclear texture descriptors ('InfoMeas', 'InvDiffMnt', and 'DiffVar'). ORA of GO cellular component terms suggested that these features are associated with spindle and microtubule formation (*Figure 6—figure supplement 3*), concomitant with altered nuclear and ER granularity. In contrast, features marking clusters enriched in KS included mitochondrial texture (InvDiffMnt), nuclear intensity, and cell shape metrics such as 'Cel_Shape_EulerNumber', and were associated with increased transcriptional activity, characteristic of cell state transitions (*Figure 6—figure supplement 3*). Altogether, these findings indicate a morphological signature associated with an established disease phenotype and showcase the value of CP in capturing new and complementary cellular phenotypes of the disease mechanisms underlying NDDs, as exemplified with KS.

## Discussion

We have performed a comprehensive characterization of iPSC-derived cortical neurons from a widely used protocol where we combined image-based readouts of cellular morphology with single-cell transcriptomics. We describe a proof-of-concept approach, where we applied CP to a heterogeneous, dynamic system – cortical neurodevelopment – and explored how the joint analysis of imaging and gene expression-based profiles of single cells can contribute to identifying new cellular phenotypes. While scRNA-seq alone allowed us to classify a diverse array of cell types produced *in vitro*, the dynamic nature of developing neurons involves alterations beyond the transcriptome such as in cellular size, shape, and complexity. In neuronal cell types, CP has previously been applied to iPSC-derived neuronal progenitors (*McDiarmid et al., 2023*) as a readout for drug treatment, rather than as a characterization approach. Application of CP in our study throughout the developmental trajectory offered novel insights on the dynamics of morphological readouts, as observed, for example, with mitochondrial channel intensity (*Figure 3b, c*).

Previous studies have attempted to link bulk gene expression and cell morphology assuming shared biological information between the two, and have found that changes in image-based features are associated with the expression of a subset of genes, often related to the cellular components and organelles stained in the assay (*Haghighi et al., 2022*; *Nassiri and McCall, 2018*). The generalizability of the CP assay (*Willis et al., 2020*) means that such direct links of gene expression to cellular components, originally observed in a human osteosarcoma cell line U2OS, are likely reproducible in different cell types. This hypothesis prompted us to link these two modalities in our neuronal system. While our sample size is small and limits our ability to generalize our findings beyond our specific set of differentiations, donor-specific variation and the large number of individual cells in our dataset boosted the capacity of the predictive model to link variation in gene expression to CP features. For example, we identified functional terms linked to CP features that recapitulate known or expected biology, such as those linked to ER, supporting our model's performance. To enhance the interpretability of novel associations between image features and their potential biological function, we applied a linear model rather than a machine learning approach, as in previous cross-modality CP-based approaches (*Way et al., 2021*).

A common challenge in modeling neurodevelopment using pluripotent stem cells, regardless of the applied protocol, lies in determining whether the cell types generated *in vitro* accurately mirror the transcriptional signatures and developmental trajectories observed *in vivo*. Previous studies in organoids have linked *in vitro*-specific cell states to aberrant oxidative or glycolytic stress (*Bhaduri et al., 2020*; *Uzquiano et al., 2022*). These alterations in energy-associated pathways are frequently associated with the limitations of the culture media to provide essential nutrients to cells, especially within the necrotic cores of organoids (*Uzquiano et al., 2022*). In a recent study (*He et al., 2024*), transcriptomic differences between cells from human neural organoids (comprising 26 protocols) and developing human brain (first-trimester) (*Braun et al., 2023*) were associated with the upregulation of canonical glycolysis and mitochondrial ATP synthesis-coupled electron transport in organoids. Similarly, we observed that low-quality cells in our 2D culture system showed distinct metabolic and energetic states compared to high-quality cells of the same type. For instance, activation of glycolysis differed among progenitors, and activation in steroids biosynthesis among excitatory neurons. A fraction of these low-quality cells could be linked to an

identity exclusively seen in organoids, most of them being either pan-neuronal or pan-radial glial. If not accounted for, mis-annotating these low-quality cells could potentially bias downstream findings in disease models.

The morphological profiling of low-quality cell types within CP data is currently not addressed in our study due to the model limitations. Replacing generic dyes with those specifically targeting neurons could improve cell type granularity (*Laber et al., 2023*), and increasing image magnification could better resolve individual organelles, potentially enhancing cell type identification by the assay. Further, to capture morphological heterogeneity, we analyzed CP readouts at a single-cell level instead of averaging CP features by well after cell segmentation as is often done, which came at the cost of increased noise (*Caicedo et al., 2022*). Although lower seeding density in culture may improve the accuracy of segmentation, we have previously observed that the viability of developing neurons is compromised in sparser culture conditions. It is also uncertain whether the 384-well format of the CP assay, as compared to 35 mm dishes used for scRNA-seq, impacts cell type production. In the future, new techniques that allow deriving CP readouts and transcriptomes from the same individual cell would offer a ground truth for model validation.

In general, a multi-modal perspective of any system offers insights that may not be visible from any one assay alone (*Way et al., 2022*). Here, CP revealed donor-specific ER changes in healthy donor lines corresponding to transcriptomic differences in ribosomal genes of inhibitory versus excitatory neurons. Understanding how interneuron production is altered at the level of both gene expression and cellular processes is key to uncovering the mechanisms implicated in NDDs, as suggested by recent findings implicating the ER and cytoskeleton in interneuron development and migration (*Meng et al., 2023*). In our model system, the 11 morphologically distinct CP clusters broadly separated progenitors from maturing neurons. In terms of cell type identification, single-cell transcriptomics remains a far more in-depth tool, particularly when cell types are closely related. However, the differences in cellular features between the various clusters suggest that CP provides information outside of the paradigm of cell types. To unravel what this biological signal might instead be capturing, we performed stratified LD score regression on the CP feature space, via the genes predicting them. While indirect, this analysis highlighted the disease relevance of morphological features in capturing heritability of brain-related traits, looking beyond cell types. Neuronal size, structure, and density have been previously shown to be impacted in various neurodevelopmental and psychiatric disorders (*Kathuria et al., 2018*; *Purcell et al., 2023*) and we show proof of concept that such changes can be captured with the CP assay.

To further validate the ability of CP to detect disease-relevant phenotypes, we applied our framework to neural progenitors derived from KS iPSC lines. Even without matched transcriptomic data, CP revealed morphologically distinct clusters enriched or depleted in KS samples, associated with altered cell cycle activity and accelerated neuronal differentiation – features previously reported in KS (*Carosso et al., 2019*) and other NDDs (*Jhanji et al., 2024*). While not performed here, such signatures could in principle be validated through orthogonal assays such as immunocytochemistry. In the future, multiplexing CP dyes with antibody-based quantification of neuronal markers could further enhance the utility of the assay in uncovering novel disease phenotypes. Nonetheless, these findings underscore the capacity of CP-derived features to capture subtle, disease-specific shifts in cell state. Moreover, linking morphological phenotypes to predicted transcriptional programs reinforced the biological relevance of these observations. Together, this highlights how image-based profiling can complement scRNA-seq by providing an orthogonal and scalable readout of developmental perturbations in both healthy and disease contexts.

In conclusion, we have performed an in-depth characterization of a widely used and highly disease-relevant cortical model using joint profiling of cell morphology and gene expression. We show that cell morphology profiling is able to capture disease-specific cellular phenotypes, complementary to those captured by scRNA-seq, and provide a novel framework for combining image-based phenotypes with transcriptomics in single cells. We highlight how joint molecular and morphological phenotyping can be widely applicable in disease modeling (*Chandrasekaran et al., 2023*; *McDiarmid et al., 2023*) and anticipate this approach to have great potential in linking genetic variants with biological processes underlying diseases.

## Materials and methods

### Human iPSC culture

The human iPSC lines HEL11.4, HEL47.2, HEL61.1, HEL61.2, HEL62.4, and HEL82.6 used in this study were acquired from the Biomedicum Stem Cell Centre (BSCC, University of Helsinki, Finland) and human iPSC lines HPSI0316i-aask_4 (aask4), HPSI0314i-hoik_1 (hoik1), HPSI0316i-ierp_4 (ierp4), HPSI0516i-oadp_4 (oadp4), HPSI0414i-oikd_2 (oikd2), and HPSI0316i-qeti_2 (qeti2) from the Human Induced Pluripotent Stem Cell Initiative were acquired from European Collection of Authenticated Cell Cultures (ECACC, Culture Collection, UK) (*Table 1*, *Supplementary file 1a*). The cells were grown on vitronectin in Essential 8 and Essential 8 Flex media (*Supplementary file 1e*) at 37°C/5% $CO_2$. iPSC maintenance in culture was performed according to HipSci guidelines: https://www.culturecollections. org.uk/media/109442/general-guidelines-for-handling-hipsci-ipscs.pdf. Cells were clump-passaged in ratios ranging from 1:4 to 1:8 using 0.5 mM EDTA diluted in DPBS−/−. Y-27632 (10 µM) was used for better cell survival at thawing. Cells were tested for mycoplasma with the MycoAlert kit, and all lines tested negative after thawing both during iPS cell culture and neural differentiation.

### Cortical neuron differentiation

The iPSC lines were differentiated into cortical progenitors and neurons using an established differentiation protocol (*Shi et al., 2012a*) with minor modifications (hereafter referred to as 'original') and a modified version from day 11 post-induction (*Supplementary file 1b*).

For neural inductions, 2–3 60–80% confluent plates of iPS cells were detached using 0.5 mM EDTA and were plated on a Matrigel plate (1:100 in DMEM/F12) in E8 medium supplemented with Y-27632 (10 µM). Dual-SMAD inhibition was initiated the following day using Neural Maintenance Medium (NMM) supplemented with SB431542 (10 µM) and LDN-193189 (200 nM) (*Supplementary file 1e*). On day 11 post-induction, cells were split in clumps in a 1:2 ratio onto laminin-coated dishes using either mechanical dissociation by gentle scraping (modified) or using Dispase II (original). The following day, the media was changed to NMM containing bFGF (20 ng/ml) for 4 days. After expansion, cells were split (1:2) with EDTA only in the modified protocol. Alternatively, a few replicates (techRep column, *Supplementary file 1b*) for days 40 and 70 were dissociated with Dispase as per the original protocol.

At day 17, the plates to be assayed at day 20 were split 1:2 as small clumps (using 1 mg/ml Dispase II or 0.5 µM EDTA for the modified protocol) onto laminin-coated plates for scRNA-seq analysis. Additionally, cells were plated in a 1:6 ratio on 24-well plates with coverslips for immunocytochemistry and in a 1:100 ratio on 384-well plates for CP. Cells assayed at days 40 and 70 were split as described in *Supplementary file 1b*.

All cells were frozen down at day 28 or 29 and thawed as per the original protocol. Cells were frozen down in a 1:1 ratio and were plated onto laminin-coated plates at thawing. Final plating for days 40 and 70 was done at day 35 when cells were passaged with Accutase. Cells were plated for scRNA-seq (1.5 million cells/35 mm dish), immunocytochemistry (75–300,000 cells/24-well plate) and CP (5000 cells/384-well plate) onto poly-L-ornithine and laminin-coated plates. Poly-L-ornithine solution was diluted to 0.01% with sterile water, coated overnight at +4°C after which the wells were washed 3 times with sterile water. Laminin was diluted in DPBS−/− and the plates were incubated at 37°C for 4 hr.

scRNA-seq samples profiled at days 20, 40, and 70 were not taken from the same continuous differentiation. Days 40 and 70 were sampled from one round of differentiation, containing 4 donors (HEL61.2, HEL11.4, HEL62.4, and HEL82.6) and with technical replicates as specified in *Supplementary file 1c*. An additional round of differentiation for the same lines was run to profile CP samples in all three time points, and in addition, day 20 scRNA-seq samples were obtained from this batch. These samples were only differentiated using the modified version of the protocol. This second batch incorporated two additional iPSC lines, HEL61.1 (clone of HEL61.2) and HEL47.2 (derived from the same donor as HEL11.4, but generated from different parental fibroblasts), requiring the barcoding technology of CellPlex to demultiplex donor identity for the day 20 scRNA-seq samples. For this time point, two independent inductions were replicated 1 week apart (batchRep column, *Supplementary file 1c*).

Finally, the CP experiment was replicated at day 20 using a third independent differentiation of the lines aask4, hoik1, ierp4, oadp4, oikd2, and qeti2 using the original version of the protocol (*Supplementary file 1b*).

## Cell preparation for scRNA-seq

Developing cortical neurons were analyzed for experiments on days 20, 40, and 70 post-neural induction. The cells were prepared for scRNA-seq as follows: The wells were washed up to three times with DPBS−/− after which they were incubated in Accutase for 5 min at 37°C. Cells were dissociated into a single-cell suspension by pipetting and added into 5 ml of 0.04% BSA in DPBS−/−. Cells were centrifuged at 180 RCF for 5 min and supernatant was removed. Cells were resuspended in 0.04% BSA and centrifuged twice more. Final resuspension of cells was done in 100 µl of 0.04% BSA, after which the cells were filtered through 40 µm FlowMe filters, followed by counting and estimation of the cell viability using Trypan Blue.

## scRNA-seq library chemistry and sequencing

Single-cell gene expression was profiled from the three time points using 10x Genomics Chromium Single Cell 3′ Gene Expression technology. Only at day 20, Cell Multiplexing technology platform (3′ CellPlex Kit) was used to demultiplex the identity of clonal cell lines. For all the time points, 10x libraries were generated using the Chromium Next GEM Single Cell 3′ Gene Expression version 3.1 Dual Index chemistry. The sample libraries were sequenced on the Illumina NovaSeq 6000 system using read lengths: 28 bp (Read 1), 10 bp (i7 Index), 10 bp (i5 Index), and 90 bp (Read 2) (*Supplementary file 1d*).

## Genotyping

To allow for donor demultiplexing during downstream analysis, iPSC lines obtained from BSCC were genotyped using SNP arrays. For this, cells were pelleted in DPBS−/− and DNA was extracted using Nucleospin DNA columns. Genotyping was performed on Illumina Global Screening Array with added GSAFIN SNPs specific for the Finnish population.

## Immunocytochemistry and image acquisition

Cells were washed three times with DPBS+/+, fixed with 4% paraformaldehyde (PFA) for 15 min followed by three DPBS washes. Cells were then permeabilized in 0.2% Triton X-100/DPBS for 15 min at RT (*Supplementary file 1e*). Coverslips were washed three times in PBST (0.1% Tween-20) followed by blocking at RT with 5% BSA/PBST for 2 hr. Cells were incubated in primary antibodies in 5% BSA/PBST overnight at 4°C (*Supplementary file 1f*). Following overnight incubation, coverslips were washed with PBST for 15 min three times, followed by incubation in secondary antibodies in 5% BSA/PBST for 1 hr. The cells were finally washed three times with DPBS for 10 min and coverslips were plated on glass slides with mounting media containing DAPI. Fixation for lines HEL62.4 and HEL82.6 was performed on day 55 rather than day 70 due to neuron detachment from the coverslips.

Imaging for day 20 ICC was performed using a Zeiss Axio Observer.Z1. The objective used was a Plan-Apochromat NA 0.8 at ×20 magnification and ×40 in batch 3 differentiations. Samples were imaged with HXP 120 V light source with the 45 Texas Red, 38HE GFP, and 49 DAPI wavelength fluorescence filters. Images were acquired using an Axiocam 506. For days 40 and 70, imaging was performed using a Zeiss Axio Imager 1 with the same objective and light source. Fluorescence filters used were 64HE mPLum, 38HE GFP, and 49 DAPI. Images were acquired using a Hamamatsu Orca Flash 4.0 LT B&W.

## CP assay and image acquisition

Cells were phenotyped using Phenovue Cell Painting Kit for 384-well plates following kit guidelines based on *Bray et al., 2016*. Cells were plated on day 17 (for day 20) or day 35 (for days 40 and 70) on PhenoPlate 384-well microplates and incubated in 37°C at 5% $CO_2$ until assay time points at days 20, 40, and 70. Staining solution 1 was added for live labeling of mitochondria (PhenoVue 641 Mitochondrial Stain) and incubated in the dark for 30 min at 37°C. Cells were fixed with 3.2% PFA for 20 min at room temperature and then were washed and incubated with 0.1% Triton X-100 followed by HBSS washes. Finally, staining solution 2 was added to the cells to label nuclei (PhenoVue Hoechst 33342 Nuclear Stain), ER (PhenoVue Fluor 488 – Concanavalin A), Golgi apparatus (PhenoVue Fluor 555 – WGA), nucleic acid (PhenoVue 512 Nucleic Acid Stain), and cytoskeleton (PhenoVue Fluor 568 – Phalloidin) and washed again with HBSS prior to imaging.

The cells were imaged with PerkinElmer Opera Phenix High Content Screening System using the Harmony Software v4.9. The imaging was done using the 40x NA 1.1 water immersion object and with the following lasers: 405 nm (emission window 435–480 nm), 488 nm (500–550 nm), 561 nm (570–630 nm), and 640 nm (650–760 nm), resulting in Golgi apparatus, nucleic acid, and cytoskeletal dyes being captured in the same channel (CGAN). Images were captured using the Andor Zyla sCMOS camera (2160 × 2160 pixels; 6.5 µm pixel size). For each well, 28 fields on 3 planes were acquired. Among images taken from multiple planes ($n$ = 3), the z-stack with the highest intensity per channel was selected for analysis.

## scRNA-seq data pre-processing and dimensionality reduction

Raw data processing and analysis were performed using 10x Genomics Cell Ranger v6.1.2 pipelines 'cellranger mkfastq' to produce FASTQ files and 'cellranger multi' to perform alignment, filtering, and UMI counting. mkfastq was run using the Illumina bcl2fastq v2.2.0 and alignment was done against human genome GRCh38. Day 20 samples were demultiplexed based on multiplexing barcode sequences using the cellranger multi pipeline. Cell recovery was 22,628 and 18,486 from the two 10x samples sequenced, with 50.89% and 73.61% cells assigned to a cell line (singlet rate), respectively, resulting in approximately 4000 cells captured per cell line across technical replicates (*Supplementary file 1d*). In 10x samples from days 40 ($n$ = 3) and 70 ($n$ = 2), donors were pooled at the time of sequencing. We assigned donor identity to each cell with *demuxlet* (*Kang et al., 2018*) by leveraging common genetic variation from the same donors previously genotyped (see Methods, Genotyping section). *Demuxlet* was run using a default prior doublet rate of 0.5. We only retained those (singleton) cells that could unambiguously be linked to a donor (average of 69.5% per pool), resulting in 5011 cells on average per donor at each time point.

scRNA-seq data was analyzed using the *Seurat* R package v4.1.1 (*Stuart et al., 2019*) using R v4.1.3. To exclude low-quality cells from analysis, we discarded cells with either less than 2000 genes expressed or more than 8000, as well as cells presenting more than 15% of reads mapping to the mitochondrial DNA. Additionally, it was ensured that each 10x sample contained between 10% and 20% reads mapping to ribosomal protein transcripts on average, as expected from neuronal populations. Following filtering, the 10x samples from all three time points were merged, and genes expressed in <0.1% of cells in the merged dataset were removed.

Gene expression counts were normalized by total expression, with a default scale factor of 10,000, and then transformed to log-space. Then, this matrix of log-normalized counts was scaled while regressing out cell cycle scores (*Tirosh et al., 2016*), depicted as the difference between G2/M and S phase scores to preserve inherent differences between cycling and non-cycling cells. Dimensionality reduction was performed via principal component analysis (PCA) using previously identified highly variable genes ($n$ = 3000). *Harmony* (v.0.1.0) was used to batch-correct the PCA embeddings from all 10x samples (*Korsunsky et al., 2019*). Based on the top 15 batch-corrected PCs, we constructed a KNN graph and clustered the cells with a resolution of 0.8, using Seurat functions *FindNeighbors()* and *FindClusters()*, respectively. We then generated a UMAP embedding again using the top 15 batch-corrected PCs from *Harmony*.

## scRNA-seq cell type annotations from *in vivo* fetal brain

The primary reference dataset used for cell type annotation was the publicly available human fetal scRNA-seq data from *Polioudakis et al., 2019*, obtained from the CoDEx online interface (http://solo.bmap.ucla.edu/shiny/webapp/). It was selected given the data and code availability, plus the metadata with regional specificity of the developing neocortex. The raw matrix of counts was log-normalized to a scale factor of 10,000 counts, and we identified the top 3,000 highly variable genes. Based on the expression of G2/M and S phase markers, we calculated cell cycle phase scores. Then, counts were scaled, regressing out 'Number_UMI', 'Library', and 'Donor' and the difference between G2/M and S cell cycle scores. Dimensionality reduction was performed via PCA on the top 3000 highly variable genes. We then batch-corrected the PCA embeddings with *Harmony* specifying the library as a covariate and used the harmonized dimensional reduction as an embedding to project our *in vitro* dataset. To transfer the cell type labels from the fetal reference to our *in vitro* query dataset, we used a two-step anchor-based approach implemented in Seurat: first running 'FindTransferAnchors' using the first 30 batch-corrected PCA from the reference embedding, and then 'MapQuery'. Further, the

tag of 'Unmapped' was assigned to cells that did not achieve a mapping score of >0.5 for any cell type label. Correlation analysis of the annotated cell types between the *in vitro* and reference datasets was performed similar to *Bhaduri et al., 2020*. On-diagonal and off-diagonal means of Pearson correlation coefficient were calculated. We classified cells in bins of low and high quality within an assigned cell type based on their best mapping score. Those cells with a score >0.5 were referred to as high quality for a predicted cell type, while cells with a score <0.5 were considered as low-quality cells. Additionally, cell types with less than 50 cells were not considered for any downstream per-cell type analyses (in this case, only ExDp2).

## Pseudotime analysis

The pseudotime trajectory was constructed using R package *monocle3* (v1.2.9). The Seurat object was converted to the *monocle3*-compatible cds object type using the function '*as.cell_data_set()*' followed by pre-processing with 100 dims, alignment by 10x sample, and clustering at a resolution of 1e−4. The '*learn_graph()*' function was used with default parameters to construct the trajectory and the cells were ordered along the trajectory using a principal node rooted in the time point day 20. Genes that vary across the trajectory were identified using '*graph_test()*' by setting the argument '*neighbor_graph*' to '*principal_graph*'. The resulting genes were grouped into modules based on '*find_gene_modules()*' after evaluating modularity using different resolution parameters {$10^{-6}$, $10^{-5}$, $10^{-4}$, $10^{-3}$, $10^{-2}$, $10^{-1}$}. Finally, the expression of these modules was aggregated per cell type using the function '*aggregate_gene_expression()*'.

## GSVA

Gene expression was aggregated per cell type (*Figures 3d, e and 5b*, *Figure 4—figure supplement 1b*) or per experimental population (*Figure 4c*) based on mean log-normalized expression values to generate pseudobulked data. GSVA was performed on pseudobulked cell types using the R package *GSVA* v1.49.4 and gene sets obtained from the Molecular Signatures Database (MSigDB) (*Liberzon et al., 2011*) compiled in the R package *msigdbr* v7.5.1. Selected pathways from KEGG (*Figures 3d–5b* and *Figure 4—figure supplement 1b*) or Gene Ontology: Cellular Components (GO CC) containing the 'MITOCHONDRIAL' term (*Figure 3e*) were tested for enrichment across cell types using the function 'gsva' with a min.sz filter of 15 and max.sz of 500. In *Figure 4—figure supplement 1e*, additional organoid data was obtained from *He et al., 2024* and subsetted for guided protocols where multiple donors were available (*Velasco et al., 2019*).

## Organoid mapping

Reference scRNA-seq data from cortical organoids was obtained from *Bhaduri et al., 2020*, via the UCSC Cell Browser (https://cells.ucsc.edu/?ds=organoidreportcard). As in the original publication, the raw count matrix was pre-processed following original filtering steps to remove cells with fewer than 500 genes expressed or with an excessive mitochondrial count fraction (>10%). Then, gene counts were normalized to a scale factor of 10,000 counts and natural-log transformed, and 3000 highly variable genes were computed. Based on the expression of G2/M and S phase markers, we calculated cell cycle scores, and the difference between G2/M and S was regressed out during data scaling. We performed PCA dimensionality reduction using the top 3000 highly variable genes. We used the first 30 PCs to project the organoid reference to our *in vitro* dataset as described for fetal mapping. This reference mapping was the second step in producing the final cell type annotation, as we only assigned the organoid cell type labels to those cells that were classified as 'unmapped' from the fetal reference-based (first-step annotation). Those cells that did not achieve a maximum mapping score of >0.5 for any cell type label in any of the two mappings were finally tagged as 'Unmapped'.

## Differential abundance (DA) testing (scRNA-seq)

We tested for DA between the cell types produced by the two versions of the protocol (original vs. modified) using *miloR* v1.4.0 (*Dann et al., 2022*). This analysis was performed solely with samples from day 40 due to the representation of all donors in each protocol version (*Supplementary file 1b*). The KNN graph was constructed using the *buildGraph()* function with 15 dimensions (*d* = 15) and 30 nearest neighbors (*k* = 30), followed by *makeNhoods()* using the same number of dimensions and sampling 20% of the graph vertices (prop = 0.2). In both cases, the dimensionality reduction from

Harmony after batch-correcting the PCA was used. To calculate the distance between neighborhoods, we used the function *calcNhoodDistance()* with 15 dimensions (*d* = 15) from Harmony. The neighborhoods were tested for DA between protocols by running *testNhoods()* with the design:

$$\sim donor + protocol$$

The resulting differentially abundant neighborhoods were annotated with cell types from the two-step annotation using *annotateNhoods*(), and neighborhoods that were not homogeneously composed of a single-cell type (fraction of cells from any given cell type <0.7) were annotated as 'Mixed'. The DA fold changes were visualized using a beeswarm plot with default significance level for Spatial FDR (<0.1).

## Gene Set Enrichment Analysis

GSEA was performed on differentially expressed genes between pan-neuronal and inhibitory cell types (*Figure 5d*) or between excitatory and inhibitory cell types (*Figure 4—figure supplement 1d*) using *fgsea* v1.20.0. The function 'fgseaMultilevel' was used on the ranked list of DEGs using KEGG gene sets from msigdbr and filtering for minsize = 10, maxsize = 500, eps = 0, and nPErmSimple = 10,000. Results were ordered by NES and filtered for pAdj < 0.05 before visualizing the top (maximum) 20 results per analysis. Additionally, in *Figures 4e and 5d*, the genes driving specific pathways were identified based on the 'leadingEdge' of each pathway and represent the core of the gene set enrichment's signal (*Mootha et al., 2003*; *Subramanian et al., 2005*). The module score for each of these core sets of genes was computed per single cell using 'AddModuleScore' from *Seurat* with default parameters.

## CP image processing

The acquired images were processed using *CellProfiler* 4.2.1 (*Stirling et al., 2021*). The workflow file is available at GitHub (CellPainting/1_primary_analysis/cellProfilerWorkflow.cppipe). Briefly, quality metrics such as blurriness and saturation were measured for each image. Nuclei were then segmented using the Hoechst channel using the minimal cross-entropy method. Then, an image summing up the channels and excluding Nucleus staining was generated to segment the cells (*Figure 1—figure supplement 2a*). Neurites were algorithmically enhanced on summed images prior to segmentation. Cell segmentation was done by propagation from the nucleus using the Otsu method. Cytoplasms were identified by subtracting nuclei area from the cell. For each object type (Cell, Nuclei, and Cytoplasm), a large number of features were measured per channel. A description of each feature can be found in the CellProfiler manual (https://cellprofiler-manual.s3.amazonaws.com/CellProfiler-4.0.4/index.html). Those features were metrics relative to channel intensity, texture, or object shape and size. To export images, the intensity of each channel was rescaled and attributed to a color. Channels merged to create one image per field in the well. An overlay of the well is then created for each field using a Python script (CellPainting/1_primary_analysis/makeComposite_stitch.py).

## CP feature quantification analysis

The feature data frames were analyzed by R v4.2 using the *oob* package (https://github.com/DimitriMeistermann/oob; *Meistermann, 2024*). First, images underwent a filtering process based on the PowerLogLogSlope, a blur metric. For each field, the average PowerLogLogSlope was calculated across the four channels. We retained images with an average value greater than –2.16, which is determined based on the distribution. Additionally, images identified as blurry by Cell Profiler are excluded. Any objects associated with discarded images are removed from the dataset. Cells exhibiting extreme values for cytoplasm area (≤1000 or ≥100,000) or too small nucleus (≤10,000) are also eliminated, as they are indicative of poor segmentation. Cells with too low nucleus channel intensity were also excluded (≤0.003), as well as being outliers in a projection with nucleus intensity as *x*-axis and sum of other channels as *y*-axis. Outliers were determined by a uniform kernel density estimation with bandwidth = 1 and removed if density ≤0.0001.

Features were then regularized to approximately follow a normal distribution. This was done by examining each feature distribution and classifying them. We defined 6 feature distribution families (*Figure 1—figure supplement 2d*, *Supplementary file 2*) and applied a specific transformation for

each. For example, intensity features underwent a $\log_2(x + 1)$ transformation. The range of each feature was then scaled to [0,1000].

The dataset, originally containing 1,186,458 cells, was reduced for ensuring a relatively balanced contribution from each experimental population (time point × cell line) and reducing the computation time. The subsampling was aimed to reach a range of 50,000–60,000 cells. This specific range was determined using bootstrapping, which involved repeatedly sampling the dataset and assessing the correlation between the subsamples and the original dataset. When 50,000 cells were selected, the average correlation was approximately 0.98. Subsampling was carried out based on experimental populations, considering differentiation day and cell line, with a maximum of 2000 cells drawn if the population exceeded this size.

Feature selection was conducted through a multi-step process. Initially, a feature graph was constructed, and edges were established between features if their correlation exceeded 0.99. Modules were subsequently identified from this graph using the function 'cluster_fast_greedy' from the *igraph* package (**Csárdi and Nepusz, 2025**). Within each of these modules, the most parsimonious feature was retained for further analysis. The features associated with spatial measures (*x–y* locations), Zernike-related values, and Cel_Neighbors_AngleBtNghbors_Adjacent were excluded from further analysis. Their signals did not exhibit a discernible pattern and, as a result, posed a potential risk of introducing noise into subsequent analyses.

Finally, a temporary cell clustering was performed (see next paragraph), and the images of cells from each cluster were assessed, revealing two clusters composed of image artifacts or dying cells. Those clusters were removed from downstream analysis. We obtained a feature matrix consisting of 223 features and 54,415 cells.

A UMAP and Leiden clustering analysis was conducted using the *oob* package with a specified parameter of 'n_neighbors = 20'. Subsequently, feature modules were identified by performing hierarchical clustering on the features, utilizing a covariance distance matrix. The number of modules was determined using the derivative loss method. This process resulted in the identification of a total of 18 modules, and they were named based on an examination of their content. To determine module activation scores, the first component of a PCA was extracted from the matrix, which contained the features of each module for all cells. The web interface for visualizing the CP dataset was coded using the d3.js framework.

## Integration of scRNA-seq and CP

For the purpose of the multi-modal analysis, scRNA-seq data was reprocessed to enhance comparability between CP features and gene expression, aiming to align these datasets as closely as possible. The used set of cells is consistent with those used in the primary scRNA-seq analyses (refer to the section on scRNA-seq data pre-processing for more details). The genes with average expression <0.005 were filtered out and normalization was performed using the computeSumFactors function from *scran* (**Lun et al., 2016**). Batches were corrected using fastMNN with $k = 5$ from the *batchelor* package (**Haghverdi et al., 2018**).

To perform the integration, only the common experimental populations (combination of differentiation day and cell line) were selected, then three metacells were created per experimental population per modality. The metacells were created by randomly attributing cells from each experimental population to one of three metacells. Feature values or gene expressions were then averaged by metacell. This led to 2 matrices of 42 metacells. These matrices were used to train lasso regression models using *sklearn* (**Pedregosa et al., 2011**) from Python (alpha = 0.02, max_iter = 10,000). For each experimental population, two metacells from each modality were used for the training and one for cross-validation. The intercepts and regression coefficients were then exported to a matrix that was used to predict CP features from gene expression. This matrix was used to build the predicted CP feature matrix of the scRNA-seq dataset. CP features markers of each CP cluster were computed in the CP feature matrix, and CP features markers of each cell type in the predicted CP feature matrix. This was done using getMarkers from the *oob* package. Subsequently, two marker score matrices were generated and correlated to obtain *Figure 2e*.

Parallel to the lasso regression models, regular linear regressions were computed with the same formula (*CP feature ~ genes*) with the aim to provide one value per gene for each CP feature. This enabled the use of GSEA to enrich CP features, using the regression coefficients as GSEA input

scores. Prior to the enrichment, a median of coefficients was computed per CP module to perform the enrichment per module with KEGG and GO databases as gene set databases.

For each cell from the scRNA-seq dataset, the predicted CP feature matrix was used along the Seurat cell cycle scores annotation to build two lasso regression models: one to predict S.score, another to predict the G2M score, with a formula on the form of *score ~ CP features*. The models were then used to predict the cell cycle score values in the CP dataset.

## AMI

Adjusted Mutual Information (AMI) was computed using the *aricode* package for R (*Sundqvist et al., 2023*). AMI is symmetric and useful for capturing complex or multi-category dependencies (*Uzquiano et al., 2022*).

## GO enrichment

ORA using GO (*Aleksander et al., 2023*) was performed using *clusterProfiler* v4.2.2 and *org.Hs.eg. db* v3.14.0. The gene universe for scRNA-seq data consists of 23,289 genes present after filtering out those expressed in <0.1% of cells (see Methods, scRNA-data pre-processing) and mapped to ENTREZ and ENSEMBL gene IDs. The *enrichGO* function from *clusterProfiler* was used to find enriched terms of all categories ('BP', 'CC', and 'MF') per previously identified gene module and top 15 terms per analysis (passing a qvalueCutoff = 0.05) were visualized using *enrichplot* v1.14.2 in *Figure 3—figure supplement 1b–f*.

GO enrichment was performed on genes predictive of CP features with the gene universe of *n* = 17,370 genes present in the linear regression matrix (from the integration of scRNA-seq and CP). Plots in *Figure 6d*, *Figure 6—figure supplement 3* correspond to the top 10 terms after filtering out redundant terms based on semantic similarity using the '*simplify*' function from *clusterProfiler* (*by* = 'p.adjust', *select_fun* = min).

## Integration of CP datasets

The raw feature matrices of the two CP datasets were merged and the feature regularization procedure (as described in CP feature quantification analysis) was applied to the reunified dataset. Harmony was used to attenuate the batch effect between the two experiments. To transfer the cluster annotation, a KNN network was computed between the two datasets, and the annotation of the CP cluster was transferred to the first neighbor of the second dataset if this neighbor had a Euclidean distance less than 400 (*Figure 6—figure supplement 2a*). An abundance table of cell lines per CP cluster was computed and analyzed with edgeR to determine the DA statistics shown in *Figure 6c*.

## Stratified LD score regression analysis

Positive markers per annotated cell type from our *in vitro* dataset were determined using *Seurat FindMarkers()* function. A 100-kb window was added on either side of each of these genes using the GenBank reference genome version NCBI:GCA_000001405.14 from GRCh37.p13 (https://ftp. ensembl.org/pub/grch37/current/fasta/homo_sapiens/pep/) and LD scores computed. We then investigated heritability enrichment for 79 traits (*Supplementary file 1g*) given our cell-type-specific annotations (*n* = 22 cell types) using stratified LD score regression implemented in LDSC (*LDSC* v1.0.0) (*Finucane et al., 2018*), with the full set of genes expressed in at least <0.1% of cells (*n* = 23,289) as the control gene set. As a comparison, we ran the same analysis using annotations for 13 brain tissues from GTEx (*GTEx Consortium, 2013*). Multiple testing correction was performed across resulting trait-specific p-values across all cell/tissue types using the Benjamini-Hochberg Procedure.

For LDSC analysis on CP feature modules, representative features were selected per feature module to avoid dilution of the signal by averaging gene coefficients across features, based on ranking criteria as follows. The features were first scored for their contribution to the module (feature loadings of PC1 from the PCA used to determine module activation scores), normalized per module. Additionally, the ratio abs(sum of coefficients)/abs(intercept) from the previously computed linear regression (CP feature ~ genes) was used as a proxy for 'predictability' of the feature, where higher value meant higher predictability. The final score was a product of the two criteria, and the top-scoring feature was selected per module. Gene markers per CP feature were selected as those >2.5 SD from the mean coefficient value for that feature, leading to between 138–270 marker genes per representative CP

feature (*n* = 18); fed to stratified LD score regression analysis as above. The same marker genes were used for GO enrichment on DA clusters in *Figure 6d*, by considering the top 3 features marking each cluster.

## Expression of brain-related developmental disorder genes

Developmental delay associated genes from the DDD study were obtained from https://www.ebi.ac.uk/gene2phenotype (version 28_7_2023). The geneset was first filtered for brain-specific genes of 'definitive' confidence. The selected (*n* = 719) genes' scaled expression was plotted per pseudobulked cell type from the fetal annotation. Then, this set was further filtered for (1) loss of function (absent gene product) genes with autosomal monoallelic requirement and (2) loss-of-function observed/expected upper bound fraction (LOEUF) score per gene <0.3, in order to only retain genes with higher probability of large consequence on cellular phenotypes. LOEUF information was downloaded from gnomAD v4 (*Karczewski et al., 2020*). The resulting *n* = 68 genes were used as the input gene set for a one-tailed Fisher's exact test per cell type or CP feature module against the background of all genes (*n* = 23,289 for scRNA-seq and 17,370 for CP) to obtain the enrichment results in *Figure 6a, b*.

## Acknowledgements

This work was primarily funded by the Helsinki Institute of Life Science (HiLIFE), University of Helsinki (UH). We also acknowledge additional financial support from the Doctoral Programme in Integrative Life Science, UH (AS), Maud Kuistila Memorial Foundation (RW), the Research Council of Finland (338835, to HK), and the Sigrid Jusélius Foundation (HK, PPC). The cell lines used in this study were obtained from the Biomedicum Stem Cell Center (BSCC), supported by HiLIFE and the Faculty of Medicine, University of Helsinki, and Biocenter Finland. Genotyping was performed at the Institute for Molecular Medicine Finland (FIMM) Genomics unit, supported by HiLIFE and Biocenter Finland. The authors would like to thank the FIMM Single-Cell Analytics unit, supported by HiLIFE and Biocenter Finland, for single-cell RNA-sequencing services, and Dr. Lassi Paavolainen and the FIMM High Content Imaging and Analysis unit for assistance with Cell Painting and for imaging and image data analysis. Imaging for ICC was performed at the Biomedicum Imaging Unit, Helsinki University, Helsinki, Finland, with the support of Biocenter Finland. The authors acknowledge Wellcome Trust Sanger Institute as the source of human induced pluripotent cell lines (healthy cohort and Kabuki syndrome) which were generated under the Human Induced Pluripotent Stem Cell Initiative funded by a grant from the Wellcome Trust and Medical Research Council, supported by the Wellcome Trust (WT098051), the NIHR/Wellcome Trust Clinical Research Facility, Manchester Centre for Genomic Medicine, Central Manchester Foundation NHS Trust, and University of Manchester, and acknowledge Life Science Technologies Corporation as the provider of Cytotune. The authors also wish to acknowledge CSC – IT Center for Science, Finland, and the Institute of Molecular Medicine Technology Center for computational resources, and Helsinki University Library for Open Access funding. Finally, the authors would like to thank Dr. Lewis Evans for providing feedback on the manuscript, and the reviewers for their constructive comments which greatly improved the manuscript.

## Additional information

### Funding

| Funder | Grant reference number | Author |
|---|---|---|
| Helsinki Institute of Life Science, Helsingin Yliopisto | | Helena Kilpinen |
| Doctoral Program in Integrative Life Science | University-funded doctoral position | Adithi Sundaresh |
| Maud Kuistila Memorial Foundation | | Rosa Woldegebriel |

| Funder | Grant reference number | Author |
|---|---|---|
| Research Council of Finland | Academy Research Fellow, #338835 | Helena Kilpinen |
| Sigrid Juselius Foundation | | Pau Puigdevall Costa Helena Kilpinen |

The funders had no role in study design, data collection, and interpretation, or the decision to submit the work for publication.

## Author contributions

Adithi Sundaresh, Formal analysis, Visualization, Methodology, Writing – original draft, Writing – review and editing; Dimitri Meistermann, Formal analysis, Visualization, Writing – original draft, Writing – review and editing; Riina Lampela, Visualization, Methodology, Writing – review and editing; Zhiyu Yang, Formal analysis, Writing – review and editing; Rosa Woldegebriel, Methodology, Writing – review and editing; Andrea Ganna, Resources; Pau Puigdevall Costa, Conceptualization, Formal analysis, Supervision, Visualization, Writing – original draft, Writing – review and editing; Helena Kilpinen, Conceptualization, Resources, Supervision, Funding acquisition, Writing – original draft, Writing – review and editing

## Author ORCIDs

Adithi Sundaresh ⬤ https://orcid.org/0000-0002-9670-8120
Dimitri Meistermann ⬤ https://orcid.org/0000-0001-6614-2325
Pau Puigdevall Costa ⬤ https://orcid.org/0000-0002-8687-4942
Helena Kilpinen ⬤ https://orcid.org/0000-0001-6692-6154

## Ethics

The cell lines used in the study are derived from human subjects. The cell lines were not established as part of the current study, but instead acquired with a material transfer agreement from the Biomedicum Stem Cell Center (BSCC) at the University of Helsinki, Finland, and the European Collection of Authenticated Cell Cultures (ECACC; lines from the Human Induced Pluripotent Stem Cell Initiative, HipSci), United Kingdom. The original ethical approvals were obtained by BSCC and the HipSci project.

## Decision letter and Author response

Decision letter https://doi.org/10.7554/eLife.102578.sa1
Author response https://doi.org/10.7554/eLife.102578.sa2

---

# Additional files

## Supplementary files

Supplementary file 1. Methods-related information. (a) Passage number information from BSCC induced pluripotent stem cell iPSC lines used in this study. (b) Protocol information for iPSC lines profiled by scRNA-seq at days 40 and 70, with their corresponding 10x library, replicate information, as well as enzyme usage for cell dissociation. (c) Cell lines included per 10x library at each differentiation day, with corresponding CellPlex barcodes (for day 20), donor and replicate information. (d) 10x libraries and sequencing information including runIDs, FASTQ IDs, and sequencing specs. (e) Reagents with their final concentration, supplier, and catalog number. (f) List of primary and secondary antibodies with their host species, dilution, supplier, and catalog numbers. (g) GWAS summary stats used for stratified LD score regression analysis.

Supplementary file 2. Cell Painting (CP) regularization. Formula of transformation applied per CP feature, ordered by feature families.

MDAR checklist

## Data availability

Raw image data from Cell Painting are deposited in the EMBL- EBI BioImage Archive under accession number S-BIAD969. Processed Cell Painting data (e.g. feature matrix, model coefficients, cell annotations) are available from Zenodo (https://zenodo.org/doi/10.5281/zenodo.10213789). Due to lack of explicit consent from the original donors, the raw scRNA-seq data cannot be shared via repositories

such as the federated European Genome-phenome Archive. Processed scRNA-seq data (raw and processed counts matrices) are currently available under restricted access from Zenodo (https://zenodo.org/doi/10.5281/zenodo.13365036). Access to these data can be requested for collaborative, academic-only research purposes from the corresponding author (helena.kilpinen@helsinki.fi). Should circumstances change, we are committed to sharing these data as openly as possible, for e.g. by making the Zenodo repository openly available. Code used to analyze the scRNA-seq and Cell Painting datasets is available at https://github.com/Kilpinen-group/cortical_diff_code/ (copy archived at *Meistermann, 2025*). Additionally, an online interface linking the CP UMAP to individual images to facilitate exploration of the CP dataset is available (see the main page of the Github repository).

The following datasets were generated:

| Author(s) | Year | Dataset title | Dataset URL | Database and Identifier |
|---|---|---|---|---|
| Sundaresh A, Meistermann D, Lampela R, Yang Z, Woldegebriel R, Ganna A, Puigdevall P, Kilpinen H | 2024 | Joint profiling of cell morphology and gene expression during in vitro neurodevelopment (scRNA-seq data) | https://doi.org/10.5281/zenodo.13365036 | Zenodo, 10.5281/zenodo.13365036 |
| Sundaresh A, Meistermann D, Lampela R, Yang Z, Woldegebriel R, Ganna A, Puigdevall P, Kilpinen H | 2025 | Joint profiling of cell morphology and gene expression during in vitro neurodevelopment (CP data) | https://doi.org/10.5281/zenodo.10213789 | Zenodo, 10.5281/zenodo.10213789 |
| Sundaresh A, Meistermann D, Lampela R, Yang Z, Woldegebriel R, Ganna A, Puigdevall P, Kilpinen H | 2023 | Characterization of cortical neurodevelopment in vitro using gene expression and morphology profiles from single cells | https://www.ebi.ac.uk/biostudies/BioImages/studies/S-BIAD969?query=S-BIAD969 | EMBL-EBI BIA, S-BIAD969 |

The following previously published dataset was used:

| Author(s) | Year | Dataset title | Dataset URL | Database and Identifier |
|---|---|---|---|---|
| Bhaduri A, Andrews MG, Mancia Leon W | 2020 | Comparison of Cortical Organoids to Human Cortex Identifies Cellular Identity and Maturation Differences | https://www.ncbi.nlm.nih.gov/search/all/?term=GSE132672 | NCBI Gene Expression Omnibus, GSE132672 |

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
