## [Editor Report]

This study presents a valuable description of human iPSC-derived neuron differentiation and highlights the utility of this cell model for investigating neurodevelopmental traits and diseases. The use of paired morphological and transcriptomic data is compelling, though the morphological data is limited in resolution and cannot distinguish cell subtypes.

---

## [Decision Letter]

**Decision letter after peer review:**

Thank you for submitting your article "Joint profiling of cell morphology and gene expression during *in vitro* neurodevelopment" for consideration by *eLife*. Your article has been reviewed by 2 peer reviewers, and the evaluation has been overseen by a Reviewing Editor and Murim Choi as the Senior Editor.

Essential Revisions (for the authors):

Although the two reviewers appreciated the importance of the study, they both think that substantial improvement is required through revision. Therefore, the editor strongly suggests that the authors should address all the major concerns raised by the reviewers.

*Reviewer #1 (Recommendations for the authors):*

In this manuscript, Sundaresh and colleagues investigated induced pluripotent stem (iPS) cell-derived cortical neurons using Cell Painting (CP) analysis, a high-content, image-based assay with single-cell resolution. Additionally, the authors combined cellular morphological profiles from CP analysis with single-cell RNA sequencing (scRNA-seq) data to expand the phenotypic readout spectrum of iPS cell-derived cortical neurons.

Using a similar approach as adopted in a previous report (Haghighi et al., 2022), the authors combined two experimental assays, CP and scRNA-seq, to predict the corresponding CP feature values from the scRNA-seq dataset. However, they found that CP largely captured only two types of cells due to low specificity and concluded that CP remains limited in its ability to resolve the detailed heterogeneity of neuronal cells. Therefore, the authors focused on mitochondrial intensity-related morphological changes by overlaying CP features onto the scRNA-seq UMAP clusters. With this focused approach, they reconfirmed previous findings (Romero-Morales et al., 2022), such as the link between progenitor cells and glycolysis, as well as the association of more mature cells with mitochondrial membrane components. This approach highlights the reciprocal interpretation of morphological changes and gene expression at the single-cell level.

In addition, the authors identified a link between reduced ER channel intensity and inhibitory neurons in one donor sample, leading to the conclusion that the major differences between inhibitory and excitatory neurons are largely driven by decreased ribosomal activity. Furthermore, the authors attempted to map the 'unmapped' cells in the scRNA-seq dataset by annotating them to cortical organoid cell types (Bhaduri et al., 2021), identifying these unmapped cells as 'pan-radial glial' or 'pan-neuronal' cell types, which are mostly intermediate transitioning cells.

Finally, the authors compared various neurological phenotypes using iPSC-derived neurons. They found a significant association between depression-related genes and excitatory neurons, as well as a link between genes related to educational attainment and inhibitory neurons. Additionally, brain-related developmental disorder genes were found to be exclusively enriched in both mature neuronal and progenitor cell types, highlighting the importance of temporal models in studying neurodevelopmental disorders.

Overall, the authors suggest that an integrative high-throughput approach, combining image-based assays with single-cell gene expression analysis, enhances the quantification of the biological processes driving neuronal differentiation *in vitro*. This study has several key merits. First, the authors demonstrated the efficiency of integrating image-based profiling with scRNA-seq data using a lasso regression model. Second, they identified that the primary differences between inhibitory and excitatory neurons are largely driven by decreased ribosomal activity, based on the enrichment of inhibitory neurons in a single donor sample.

However, the results reported by the authors here have limitations. Although they attempted to apply similar methods as described in Haghighi et al. (2022) in their neuronal culture model, it is disappointing that Cell Painting (CP) was able to capture only two broad subclusters due to low specificity. This limitation ultimately forced the authors to focus on mitochondria-associated features and unmapped cells from the scRNA-seq dataset (Figure 5). Second, while it is intriguing that reduced ribosomal activity in pseudo-time analysis appears to be a major factor inducing differences between excitatory and inhibitory neurons, this finding is limited to a single donor sample and does not seem statistically significant (Figure 4e). Furthermore, there is a lack of direct or indirect evidence to support how reduced ribosomal activity leads to the production of inhibitory neurons. Third, the authors suggested that approximately 45% of Deciphering Developmental Disorder (DDD) genes are expressed in mature neuronal cell types. However, it also appears that many other DDD genes are enriched in non-neuronal cells, such as pericytes and oligodendrocyte progenitor cells.

Concerns and limitations:

This is an interesting and relevant model; however, it builds upon previous ones (Haghighi et al. 2022). The reviewer's enthusiasm is somewhat tempered by the study's technological focus and the lack of proof-of-concept demonstrating the model's capacity to facilitate discoveries. The authors state, "We provide a novel framework for combining image-based phenotypes with transcriptomics in single cells and highlight the potential of cell morphology measurements to capture dynamic cell states and biological processes that are complementary to those captured by scRNA-seq," in the Discussion section. This raises the question: why should we employ both methods simultaneously to understand cell dynamics even CP analysis captures only two subtypes and scRNA-seq could offers a much higher complexity of cell type characterization? What insights can we gain from this model? Additionally, the authors could have elaborated more on the functional aspects of neuronal differentiation. While they focus on the transcriptional state of neuronal differentiation and show that it resembles fetal brain cells, which is impressive, it is also somewhat expected. What knowledge can we derive from this model?

*Reviewer #2 (Recommendations for the authors):*

Summary

The authors aimed to characterize the differentiation process of iPSC-derived cortical neurons using a combination of Cell Painting (CP), a high-content imaging approach, and single-cell RNA sequencing. They sought to identify the relationship between neuronal morphology and gene expression over time, and to assess the utility of these methods in modeling neurodevelopmental processes and related disorders.

Strengths:

1. Methodology: Their multi-modal approach, integrating both CP and scRNA-seq, provides a detailed portrait of their cell culture system. Established analytical methods are used effectively, and novel methods are well-described and appear to be sound.

2. Temporal Analysis: Data collection at three distinct time points gives a dynamic view of neurodevelopmental processes.

3. Disease Modeling Potential: Identification of cell types relevant to brain-related traits such as schizophrenia and bipolar disorder highlights the potential utility of iPSC-derived neurons for disease modeling.

Weaknesses:

1. Cell Type Resolution: While CP distinguished broad cell categories, it struggled to resolve finer cell types compared to scRNA-seq, suggesting limitations in the current assay.

2. Donor-Specific Effects: Strong variability between donors was observed, which, while informative, may complicate broader generalization of their results.

3. Limitations of the Cell Model: The differentiation protocol yields a variable population of cell types/states, and the cells enriched for disease heritability make up less than half of the total, even after months in culture. This may limit the usefulness of the culture system for disease modeling.

Appraisal of Aims and Conclusions

The authors aimed to use joint profiling of cell morphology and gene expression to enhance the understanding of neuronal differentiation and identify disease-relevant cell types. This goal was largely achieved, as the study provided detailed insights into the differentiation trajectory, identified key transcriptional markers, and demonstrated the relevance of iPSC-derived neurons for studying neurodevelopmental disorders. New methods developed to integrate CP and transcriptional data were particularly interesting.

However, there are a number of "negative" results in this study: limited insights were gained from the CP assay, a large degree of variation was observed between donor lines, and only 60% of the cells in their culture system are fully differentiated toward a recognizable type. These are important observations for the field to grapple with, and the authors are commended for presenting these results in a relatively clear manner. Their data and integration methods are likely to be valuable resources for efforts to improve differentiation protocols and morphological assays.

In summary, this study offers a valuable contribution by demonstrating the potential power of combining high-content imaging and transcriptomics. The methodologies and insights presented are likely to inspire further studies and applications in disease modeling.

1. Clarify interpretation of Cell Paint and scRNA-seq integration.

The cell paint comparison to the scRNA-seq seems to reveal two populations (Figure 2e): high mito intensity (progenitors, glia, unmapped) and low mito intensity (neurons). But then this thread is complicated by the projection of the CP onto the scRNA-seq UMAP in Figure 3c, which (perhaps inaccurately?) shows MGE as high mito – in a later result, they do not have high GSVA enrichment for mitochondrial pathways. Can the authors please explain this apparent discrepancy? Relatedly, are the circles in 3C meant to highlight the same area on both plots? If this can be simplified to high and low, please consider making this revision.

2. Suggest toning down emphasis on the complementarity of the two modalities: the data imply that the cell paint is simply a lower resolution picture.

3. Test discussion point about inclusion/exclusion of low-quality cells on bias in downstream findings (p12, line 18-20).

The authors do a nice job annotating cells and describing potential issues with model heterogeneity, and warn that this level of annotation is required for downstream interpretation. It would be helpful to demonstrate this with an example. The authors should consider including a comparison of sLDSC results depending on if they had included the low quality cells to the closest annotated cluster vs. not. Are enrichments reduced?

4. Suggest changing the color panel for the clusters – they are very hard to distinguish.

[Editors' note: further revisions were suggested prior to acceptance, as described below.]

Thank you for resubmitting your work entitled "Joint profiling of cell morphology and gene expression during *in vitro* neurodevelopment" for further consideration by *eLife*. Your revised article has been evaluated by Murim Choi (Senior Editor) and a Reviewing Editor.

The manuscript has been improved but there are some remaining issues that need to be addressed, as outlined below:

One of the reviewers provided the following comment, suggesting additional experiment. However, as a BRE, I would override the comment and merely ask for editing text to incorporate the reviewer's suggestion:

The authors analyzed CP in KS iPSC-derived neurons and presented their results as follows:

"Differential abundance (DA) analysis between KS and healthy donor samples, stratified by the transferred cluster labels, revealed clusters that were significantly enriched or depleted in KS samples. Specifically, the clusters 'd20.endoRetNeg' and 'd20.nucNeg' were differentially abundant in healthy donor lines across both batches (Figure 6c); notably, 'd20.nucNeg' was previously shown to be associated with increased cell cycle activity (Figures 1c, 2c). On the other hand, the clusters upregulated in KS lines ('d40.early' and 'd40.smallCells') represented d40-like cell states, despite being derived from day 20 differentiations (Figure 6c)."

However, validation evidence, such as immunostaining, is currently lacking. For example, if CP features like "d40.early" and "d40.smallCells" were indeed upregulated in the KS line, experimental validation and corresponding quantification data should be added to the supplementary material. (i.e., validation for premature differentiation in KS neurons). This addition would significantly increase the reliability of the current approach.

*Reviewer #1 (Recommendations for the authors):*

In this revision, the authors conducted additional experiments to address the reviewers' questions. The authors have addressed my concerns by providing new data were derived from the differentiation of six additional cell lines, representing two healthy individuals and four individuals with Kabuki Syndrome as well as further clarification. With new supporting data and clarification, this study/manuscript is much better improved.

The authors analyzed CP in KS iPSC-derived neurons and presented their results as follows:

"Differential abundance (DA) analysis between KS and healthy donor samples, stratified by the transferred cluster labels, revealed clusters that were significantly enriched or depleted in KS samples. Specifically, the clusters 'd20.endoRetNeg' and 'd20.nucNeg' were differentially abundant in healthy donor lines across both batches (Figure 6c); notably, 'd20.nucNeg' was previously shown to be associated with increased cell cycle activity (Figures 1c, 2c). On the other hand, the clusters upregulated in KS lines ('d40.early' and 'd40.smallCells') represented d40-like cell states, despite being derived from day 20 differentiations (Figure 6c)."

However, validation evidence, such as immunostaining, is currently lacking. For example, if CP features like "d40.early" and "d40.smallCells" were indeed upregulated in the KS line, experimental validation and corresponding quantification data should be added to the supplementary material. (i.e., validation for premature differentiation in KS neurons). This addition would significantly increase the reliability of the current approach.

---

## [Author Response]

Essential Revisions (for the authors):Although the two reviewers appreciated the importance of the study, they both think that substantial improvement is required through revision. Therefore, the editor strongly suggests that the authors should address all the major concerns raised by the reviewers.Reviewer #1 (Recommendations for the authors):In this manuscript, Sundaresh and colleagues investigated induced pluripotent stem (iPS) cell-derived cortical neurons using Cell Painting (CP) analysis, a high-content, image-based assay with single-cell resolution. Additionally, the authors combined cellular morphological profiles from CP analysis with single-cell RNA sequencing (scRNA-seq) data to expand the phenotypic readout spectrum of iPS cell-derived cortical neurons.Using a similar approach as adopted in a previous report (Haghighi et al., 2022), the authors combined two experimental assays, CP and scRNA-seq, to predict the corresponding CP feature values from the scRNA-seq dataset. However, they found that CP largely captured only two types of cells due to low specificity and concluded that CP remains limited in its ability to resolve the detailed heterogeneity of neuronal cells. Therefore, the authors focused on mitochondrial intensity-related morphological changes by overlaying CP features onto the scRNA-seq UMAP clusters. With this focused approach, they reconfirmed previous findings (Romero-Morales et al., 2022), such as the link between progenitor cells and glycolysis, as well as the association of more mature cells with mitochondrial membrane components. This approach highlights the reciprocal interpretation of morphological changes and gene expression at the single-cell level.In addition, the authors identified a link between reduced ER channel intensity and inhibitory neurons in one donor sample, leading to the conclusion that the major differences between inhibitory and excitatory neurons are largely driven by decreased ribosomal activity. Furthermore, the authors attempted to map the 'unmapped' cells in the scRNA-seq dataset by annotating them to cortical organoid cell types (Bhaduri et al., 2021), identifying these unmapped cells as 'pan-radial glial' or 'pan-neuronal' cell types, which are mostly intermediate transitioning cells.Finally, the authors compared various neurological phenotypes using iPSC-derived neurons. They found a significant association between depression-related genes and excitatory neurons, as well as a link between genes related to educational attainment and inhibitory neurons. Additionally, brain-related developmental disorder genes were found to be exclusively enriched in both mature neuronal and progenitor cell types, highlighting the importance of temporal models in studying neurodevelopmental disorders.Overall, the authors suggest that an integrative high-throughput approach, combining image-based assays with single-cell gene expression analysis, enhances the quantification of the biological processes driving neuronal differentiation *in vitro*. This study has several key merits. First, the authors demonstrated the efficiency of integrating image-based profiling with scRNA-seq data using a lasso regression model. Second, they identified that the primary differences between inhibitory and excitatory neurons are largely driven by decreased ribosomal activity, based on the enrichment of inhibitory neurons in a single donor sample.However, the results reported by the authors here have limitations. Although they attempted to apply similar methods as described in Haghighi et al. (2022) in their neuronal culture model, it is disappointing that Cell Painting (CP) was able to capture only two broad subclusters due to low specificity. This limitation ultimately forced the authors to focus on mitochondria-associated features and unmapped cells from the scRNA-seq dataset (Figure 5).

We thank the reviewer for their careful consideration of our manuscript and for drawing our attention to the phrasing involved in discussing the seemingly limited resolution of Cell Painting in characterizing cell type identity. While it is true that in our dataset, the overlapping signal between CP and scRNA-Seq can primarily distinguish ‘only’ between progenitor and neuronal populations, it can be argued that this in itself is a remarkable achievement from a highly generalized assay, such as CP, particularly in a developmental and temporal system where differences among cell types are still emerging. Our data suggests that the variability in the morphological readouts capture primarily variability beyond cell types. Indeed, this is one of the goals of our study – to explore what the assay is capturing, if not cell types as defined by the transcriptome. We have taken this opportunity to add both new analyses and data to highlight the granularity of the Cell Painting feature space, as well as to uncover the potential biological meaning of this granularity.

Second, while it is intriguing that reduced ribosomal activity in pseudo-time analysis appears to be a major factor inducing differences between excitatory and inhibitory neurons, this finding is limited to a single donor sample and does not seem statistically significant (Figure 4e). Furthermore, there is a lack of direct or indirect evidence to support how reduced ribosomal activity leads to the production of inhibitory neurons.

We thank the reviewer for pointing this out. The link between inhibitory neuron production and ribosomal effects was intended to showcase an example where Cell Painting could pick up a cell line-specific phenotype that could be verified with scRNA-seq (an inherently more interpretable assay). However, we agree with the reviewer that an observation from a single donor is anecdotal and therefore, we have now replicated this observation in two publicly available *in vitro* organoid datasets^1,2^. We report a moderate negative association between inhibitory neuron proportion and ribosomal activity, which we present in the revised version of the manuscript. Additionally, we find that this association is completely absent when comparing excitatory neurons in the same datasets (r=-0.38, p=0.03). We agree that this does not provide direct causal evidence, but our aim was instead to showcase the application of CP to identify donor-specific phenotypes, and to support this finding from the transcriptomic standpoint. Demonstrating whether or not reduced ribosomal activity and the production of inhibitory neurons are causally linked falls beyond the scope of this manuscript. We have updated the text to clarify this.

Third, the authors suggested that approximately 45% of Deciphering Developmental Disorder (DDD) genes are expressed in mature neuronal cell types. However, it also appears that many other DDD genes are enriched in non-neuronal cells, such as pericytes and oligodendrocyte progenitor cells.

DDD genes include a large fraction of genes that have highly generalized functions, such as chromatin remodeling^3^, and are expected to be expressed broadly in many, if not most, cell types. Why mutation effects manifest only in some cell types and tissues remains poorly understood. We have used a differentiation system that is capable of producing multiple cell types to show that some DDD genes are better captured in progenitor-like cells and others in neuron-like cells, as defined by their expression level. However, it may be that studying non-neuronal cells, such as pericytes, will also be relevant to understanding the disease mechanisms of NDDs.

Concerns and limitations:This is an interesting and relevant model; however, it builds upon previous ones (Haghighi et al. 2022). The reviewer's enthusiasm is somewhat tempered by the study's technological focus and the lack of proof-of-concept demonstrating the model's capacity to facilitate discoveries. The authors state, "We provide a novel framework for combining image-based phenotypes with transcriptomics in single cells and highlight the potential of cell morphology measurements to capture dynamic cell states and biological processes that are complementary to those captured by scRNA-seq," in the Discussion section. This raises the question: why should we employ both methods simultaneously to understand cell dynamics even CP analysis captures only two subtypes and scRNA-seq could offers a much higher complexity of cell type characterization? What insights can we gain from this model?

We thank the reviewer for their careful consideration of our manuscript and for the highly useful comments.

The readouts provided by scRNA-seq and Cell Painting differ substantially and capture distinctive cellular properties at markedly different resolutions. For example, mRNA levels are only partially correlated with protein levels^6^ and thus, to downstream biological processes, whereas cell morphology reflects cell function more directly. Thus, our motivation to use Cell Painting was to collect phenotypic readouts beyond gene expression – not simply to evaluate its ability to recapitulate what is seen with scRNA-seq. We recognize that upon comparison to cell types defined by transcriptomics, the resolution of Cell Painting appears low, however, the granularity of unsupervised clustering in the CP data suggests that it captures something other than cell types, which we set out to explore.

Image-based phenotypic features frequently lack direct interpretability. A joint analysis with transcriptomic data, as proposed in Haghighi *et al.*, allows building links between image features and genes. This enables linking cell morphology phenotypes with genes, adding a layer of interpretability for e.g. for disease-specific studies. In our manuscript, we have notably expanded the approach in Haghighi *et al. –* specifically, we applied the predictive model in the framework of single cells and of a time course experiment, with high cell type heterogeneity. This allowed us to link morphology phenotypes not only with genes, but cell types and developmental stages.

To better demonstrate the biological insights that can be learnt with this approach, we have included additional CP data into the manuscript. Specifically, we collected CP data from 12,000 day 20 neurons following the same approach as in the original manuscript. These data were derived from the differentiation of six additional cell lines (Table 1), representing two healthy individuals and four individuals with Kabuki Syndrome (KS), a rare neurodevelopmental disorder. We integrated the new CP data with the day 20 of the initial CP dataset (Figure 6—figure supplement 2) and, overall, performed a differential abundance test of CP clusters between the cases (n=4) and the controls (n=8). By first looking at CP data alone, we detected a shift in neuronal maturity in the Kabuki samples compared to the controls, as indicated by overrepresentation of day 40-like cells compared to WT at the same time point. This was linked with differential activation of nuclear texture-related features in WTs versus features marking mitochondrial texture and cell shape in the KS lines (Figures 6c, 1c, Figure 1—figure supplement 3b). We then applied the predictive model which we originally presented (based on WT lines from differentiation batch 1 only) to the additional dataset, allowing us to link the morphological changes associated with the KS cell lines to functional enrichment terms. We found that KS lines showed an overall upregulation of patterning and differentiation activity, and downregulation of terms associated with cell cycle (Figure 6d, Figure 6—figure supplement 3), suggesting a precocious differentiation phenotype. With this approach, we demonstrate the utility of CP in identifying disease-relevant cellular phenotypes, and highlight the relevance of the predictive model in jointly deciphering the expected functional impacts of these changes. Additionally, the integration of the new dataset with the original, along with the transfer of initial cluster labels, demonstrates that the original clustering was robust enough to enable biologically meaningful insights even across independent experiments with distinct donor backgrounds.

We have updated the manuscript to include this new analysis and addressed each of the major points throughout the manuscript. The new data and results have been incorporated into Figure 6 (new panels b, c, and d) and Figure 6—figure supplements 2 and 3.

Additionally, the authors could have elaborated more on the functional aspects of neuronal differentiation. While they focus on the transcriptional state of neuronal differentiation and show that it resembles fetal brain cells, which is impressive, it is also somewhat expected. What knowledge can we derive from this model?

We agree with the reviewer that the resemblance of *in vitro* neurons to fetal brain cells is indeed an expected, and desired, result. However, the aim of the study was not so much to describe a novel differentiation system but instead validate and characterise it on the single cell level, to inform subsequent studies. While differentiation protocols have an expected outcome, it is not always known if a) the outcome is achieved and b) what other cell types might be present in the dish during differentiation. We attempt here to show both the strengths and limitations of a commonly used *in vitro* system to recapitulate corticogenesis, as well as highlight its disease relevance. To this end, in the revised manuscript, we have performed stratified LD score regression analysis not only with scRNA-seq, but also with Cell Painting feature modules (Figure 6b). We found that the CP feature modules ‘integrInt.cytoStruct’ and ‘Nuc_shape_Eccentricity’ were significantly enriched (p<0.05) for variation associated with ADHD and Schizophrenia, respectively, showing that both *in vitro* cell types and morphological cell states are relevant for capturing heritability of brain-related disease traits.

Reviewer #2 (Recommendations for the authors):SummaryThe authors aimed to characterize the differentiation process of iPSC-derived cortical neurons using a combination of Cell Painting (CP), a high-content imaging approach, and single-cell RNA sequencing. They sought to identify the relationship between neuronal morphology and gene expression over time, and to assess the utility of these methods in modeling neurodevelopmental processes and related disorders.Strengths:1. Methodology: Their multi-modal approach, integrating both CP and scRNA-seq, provides a detailed portrait of their cell culture system. Established analytical methods are used effectively, and novel methods are well-described and appear to be sound.2. Temporal Analysis: Data collection at three distinct time points gives a dynamic view of neurodevelopmental processes.3. Disease Modeling Potential: Identification of cell types relevant to brain-related traits such as schizophrenia and bipolar disorder highlights the potential utility of iPSC-derived neurons for disease modeling.Weaknesses:1. Cell Type Resolution: While CP distinguished broad cell categories, it struggled to resolve finer cell types compared to scRNA-seq, suggesting limitations in the current assay.

We thank the reviewer for their insightful comments, which have helped us to improve the overall quality and relevance of the manuscript. This particular concern was also raised by Reviewer #1, which we addressed in detail earlier in our response (p.3).

2. Donor-Specific Effects: Strong variability between donors was observed, which, while informative, may complicate broader generalization of their results.

Donor variability is a common feature of iPSC-based models^4^. Given this, we agree with the reviewer that it may not be possible to fully generalize all results from this study to others. However, in the revised version of the manuscript, we present two observations that support the notion that the results from the original dataset are not unique to our study. First, we replicate the link between inhibitory neuron production and ribosomal effects, presented in the original manuscript and observed in a single donor (Figure 4c,e), in two other publicly available *in vitro* neuron datasets^1,2^ (five additional cell lines). Please also see response to Reviewer #1 concerning this specific result (p.3). Second, to address the generalization of results in CP data, we integrated a second independent experiment consisting of distinct donors in the revised version of the manuscript.

3. Limitations of the Cell Model: The differentiation protocol yields a variable population of cell types/states, and the cells enriched for disease heritability make up less than half of the total, even after months in culture. This may limit the usefulness of the culture system for disease modeling.

iPSC differentiation protocols can be approximated as less directed morphogen-based methods and more directed induced methods, of which the latter ones produce cultures strongly enriched for a particular cell type. In contrast, methods using combinations of morphogens, such as applied in this study, have more physiological resemblance and generate heterogeneous cultures with diverse cell type representation over a longer period of time. Therefore, if only a subset of the cell types are disease-relevant, their proportion in culture will not be as high as with induced methods, and the utility for disease modelling depends on the desired readout. Both approaches have their merits and limitations – for example although induced approaches may produce more enriched cell types, it has been shown that circumventing progenitor states in these approaches can abolish NDD-relevant phenotypes^5^.

Here, we chose this differentiation approach to explore the disease-relevance of different neuronal cell types. Once the most relevant cell types have been established in specific diseases, future studies may choose to apply protocols that enrich that particular cell type. It is also worth noting that enrichment for disease heritability can only be observed for traits for which sufficiently powered GWAS summary statistics are available.

Appraisal of Aims and ConclusionsThe authors aimed to use joint profiling of cell morphology and gene expression to enhance the understanding of neuronal differentiation and identify disease-relevant cell types. This goal was largely achieved, as the study provided detailed insights into the differentiation trajectory, identified key transcriptional markers, and demonstrated the relevance of iPSC-derived neurons for studying neurodevelopmental disorders. New methods developed to integrate CP and transcriptional data were particularly interesting.However, there are a number of "negative" results in this study: limited insights were gained from the CP assay, a large degree of variation was observed between donor lines, and only 60% of the cells in their culture system are fully differentiated toward a recognizable type. These are important observations for the field to grapple with, and the authors are commended for presenting these results in a relatively clear manner. Their data and integration methods are likely to be valuable resources for efforts to improve differentiation protocols and morphological assays.In summary, this study offers a valuable contribution by demonstrating the potential power of combining high-content imaging and transcriptomics. The methodologies and insights presented are likely to inspire further studies and applications in disease modeling.

We thank the reviewer for their careful consideration of our manuscript and for appreciating also the ‘negative’ results of the study.

1. Clarify interpretation of Cell Paint and scRNA-seq integrationThe cell paint comparison to the scRNA-seq seems to reveal two populations (Figure 2e): high mito intensity (progenitors, glia, unmapped) and low mito intensity (neurons). But then this thread is complicated by the projection of the CP onto the scRNA-seq UMAP in Figure 3c, which (perhaps inaccurately?) shows MGE as high mito – in a later result, they do not have high GSVA enrichment for mitochondrial pathways. Can the authors please explain this apparent discrepancy? Relatedly, are the circles in 3C meant to highlight the same area on both plots? If this can be simplified to high and low, please consider making this revision.

Thank you for highlighting the apparent discrepancy between the predicted high mitochondrial intensity in MGE cells (Figure 3c) and their low GSVA enrichment for mitochondrial pathways. We would like to clarify that the Cell Painting model was trained in an unsupervised manner. Therefore, the predicted intensities for cellular components—such as mitochondria—reflect morphological patterns statistically associated with the expression of pathways, rather than direct functional readouts of those pathways.

In this specific case, further analysis revealed that the gene *PEG10* plays a key role in driving this discrepancy. *PEG10* is strongly associated with high predicted mitochondrial intensity across conditions, likely due to co-expression patterns captured by the model. At the same time, *PEG10* is also amongst the genes differentially expressed in InMGE cells within our dataset. However, it is not functionally involved in mitochondrial pathways, which explains why GSVA enrichment for mitochondrial functions remains low in InMGE cells despite the elevated predicted mitochondrial intensity. Although there are better markers of InMGE cells in our data in the transcriptomic space (Figure 4—figure supplement 1a), only genes that are both differentially expressed across cell types and predictive of CP-derived morphological features influence the model’s projections.

The captured co-expression patterns could instead be explained by the fact that the CP cluster marked by the lowest ribosomal association (likely containing low-ribosome inhibitory neurons), groups amongst progenitors. We thus hypothesize that the interneurons in our dataset are still immature given the early differentiation time point, which would explain their partial morphological clustering with progenitors, and vice versa, their apparent high mitochondrial intensity when these CP features are projected onto the scRNA-seq UMAP.

We appreciate the reviewer having noticed this point and have revised the text to make this clearer as well as removed the circles in Figure 3c to avoid any misinterpretation.

2. Suggest toning down emphasis on the complementarity of the two modalities: the data imply that the cell paint is simply a lower resolution picture.

We appreciate the reviewer’s input regarding the resolution offered by Cell Painting, and recognize that the text alludes to the assay being ‘low-resolution’. We would like to clarify that while the standard CP assay is unable to capture cell type resolution to the extent of transcriptomic readouts, we find that it provides useful and unbiased information (at high throughput and low cost) about the morphology of the developing cells (such as size, density, shape, and mitochondrial readouts), as highlighted throughout the manuscript. However, we believe that the field currently lacks the means to fully decipher and use the data captured by CP – for example, interpreting what morphologically distinct clusters represent, if not cell types. For this purpose, we have built on the functional enrichment analysis of CP features included in the original manuscript (Figure 1c) by including them in a new analysis of disease heritability captured by CP feature module-associated genes (Figure 6b). The main text has been updated to highlight these points. We believe that as the single cell field moves from rigid ‘cell type’ definitions towards a more nuanced understanding of ‘cell states’, morphological data such as that offered by CP can be better utilized and interpreted.

Please also see the earlier responses to Reviewer #1 (p.6,7), who raised a similar point.

3. Test discussion point about inclusion/exclusion of low-quality cells on bias in downstream findings (p12, line 18-20).

We thank the reviewer for this suggestion. Upon performing sLDSC (as in Figure 6a) using a version of the annotation where low quality cells are included, we find that the enrichment signal is diluted, albeit not greatly (see Author response image 1). We observed that the low-quality cells in our dataset still expressed the canonical markers of the closest-mapped cell type, albeit at lower levels (Figure 5—figure supplement 1a) and thus, a strong dilution of the signal in a marker based analysis was unlikely. Our findings indicate that these cells instead differ metabolically from their high-quality counterparts (Figure 5b) and suggest this should be accounted for in transcriptomics-based disease studies *in vitro* to ensure that cells from all sample groups are of comparable quality.

**Author response image 1. sa2fig1:** Stratified LD score regression analysis including low-quality cells. Results of the stratified LD score regression analysis shown for selected brain-related traits per cell type from our *in vitro* differentiation, prior to filtering for reference mapping thresholds. The adjusted p-values are reduced in comparison with the analysis shown in original Figure 6a, indicating a dilution of signal for the same set of common diseases. Tile color represents the corresponding p-values after multiple testing correction across traits and cell types, with the significance level indicated as follows: pAdj<0.05(*), pAdj<0.01(**) and pAdj<0.001 (***).

4. Suggest changing the color panel for the clusters – they are very hard to distinguish.

We agree with the reviewer that the current colors make the clusters hard to distinguish. We have updated Figures 2a, 2d and 4e with a new color scheme. Additionally, for Figure 2a, we have included a faceted plot in the supplement (Figure 2—figure supplement 2).

References

He, Z. *et al.* An integrated transcriptomic cell atlas of human neural organoids. *Nature* 635,

690–698 (2024).

Bhaduri, A. *et al.* Cell stress in cortical organoids impairs molecular subtype specification.

*Nature* 578, 142–148 (2020).

Mossink, B., Negwer, M., Schubert, D. & Nadif Kasri, N. The emerging role of chromatin remodelers in neurodevelopmental disorders: a developmental perspective. *Cell. Mol. Life Sci.* 78, 2517–2563 (2021).

Jerber, J. *et al.* Population-scale single-cell RNA-seq profiling across dopaminergic neuron differentiation. *Nat. Genet.* 53, 304 (2021).

Schafer, S. T. *et al.* Pathological priming causes developmental gene network

heterochronicity in autistic subject-derived neurons. *Nat. Neurosci.* 22, 243–255 (2019).

Buccitelli, C. & Selbach, M. mRNAs, proteins and the emerging principles of gene expression control. *Nat. Rev. Genet.* 21, 630–644 (2020).

[Editors’ note: what follows is the authors’ response to the second round of review.]

The manuscript has been improved but there are some remaining issues that need to be addressed, as outlined below:One of the reviewers provided the following comment, suggesting additional experiment. However, as a BRE, I would override the comment and merely ask for editing text to incorporate the reviewer's suggestion:The authors analyzed CP in KS iPSC-derived neurons and presented their results as follows:"Differential abundance (DA) analysis between KS and healthy donor samples, stratified by the transferred cluster labels, revealed clusters that were significantly enriched or depleted in KS samples. Specifically, the clusters 'd20.endoRetNeg' and 'd20.nucNeg' were differentially abundant in healthy donor lines across both batches (Figure 6c); notably, 'd20.nucNeg' was previously shown to be associated with increased cell cycle activity (Figures 1c, 2c). On the other hand, the clusters upregulated in KS lines ('d40.early' and 'd40.smallCells') represented d40-like cell states, despite being derived from day 20 differentiations (Figure 6c)."However, validation evidence, such as immunostaining, is currently lacking. For example, if CP features like "d40.early" and "d40.smallCells" were indeed upregulated in the KS line, experimental validation and corresponding quantification data should be added to the supplementary material. (i.e., validation for premature differentiation in KS neurons). This addition would significantly increase the reliability of the current approach.Reviewer #1 (Recommendations for the authors):In this revision, the authors conducted additional experiments to address the reviewers' questions. The authors have addressed my concerns by providing new data were derived from the differentiation of six additional cell lines, representing two healthy individuals and four individuals with Kabuki Syndrome as well as further clarification. With new supporting data and clarification, this study/manuscript is much better improved.The authors analyzed CP in KS iPSC-derived neurons and presented their results as follows:"Differential abundance (DA) analysis between KS and healthy donor samples, stratified by the transferred cluster labels, revealed clusters that were significantly enriched or depleted in KS samples. Specifically, the clusters 'd20.endoRetNeg' and 'd20.nucNeg' were differentially abundant in healthy donor lines across both batches (Figure 6c); notably, 'd20.nucNeg' was previously shown to be associated with increased cell cycle activity (Figures 1c, 2c). On the other hand, the clusters upregulated in KS lines ('d40.early' and 'd40.smallCells') represented d40-like cell states, despite being derived from day 20 differentiations (Figure 6c)."However, validation evidence, such as immunostaining, is currently lacking. For example, if CP features like "d40.early" and "d40.smallCells" were indeed upregulated in the KS line, experimental validation and corresponding quantification data should be added to the supplementary material. (i.e., validation for premature differentiation in KS neurons). This addition would significantly increase the reliability of the current approach.

In line with the recommendation from Reviewer #1 and yourself, we have added the following text to the Discussion of the manuscript:

“Even without matched transcriptomic data, CP revealed morphologically distinct clusters enriched or depleted in KS samples, associated with altered cell cycle activity and accelerated neuronal differentiation — features previously reported in KS (Carosso et al. 2019) and other NDDs (Jhanji et al. 2024). While not performed here, such signatures could in principle be validated through orthogonal assays such as immunocytochemistry. In the future, multiplexing CP dyes with antibody-based quantification of neuronal markers could further enhance the utility of the assay in uncovering novel disease phenotypes. Nonetheless, this study underscores the capacity of CP-derived features to capture subtle, disease-specific shifts in cell state. Moreover, linking morphological phenotypes to predicted transcriptional programs reinforced the biological relevance of these observations. Together, this highlights how image-based profiling can complement scRNA-seq by providing an orthogonal and scalable readout of developmental perturbations in both healthy and disease contexts.” (page 13 of the revised manuscript)